

# The rising productivity of alpine grassland under warming, drought and N-deposition treatments

Matthias Volk[1], Matthias Suter[2], Anne-Lena Wahl[1], Seraina Bassin[1,3]

[1]Climate and Agriculture, Agroscope, Reckenholzstrasse 191, 8046 Zurich, Switzerland

[2]Forage Production and Grassland Systems, Agroscope, Reckenholzstrasse 191, 8046 Zurich, Switzerland

[3]Pädagogische Hochschule Schaffhausen, Ebnatstrasse 80, 8200 Schaffhausen, Switzerland

Corresponding author: Matthias Volk (matthias.volk@agroscope.admin.ch)



**Abstract**
We conducted a four-year warming × moisture × N-deposition field-experiment (AlpGrass) with 216 turf
monoliths from six different subalpine pastures (sites of origin). At a common location, the monoliths were
replanted at six climate scenario sites (CS) along an altitudinal gradient from 2360 to 1680 m a.s.l., representing
an April - October temperature change of -1.4 °C to +3.0 °C, compared to $CS_{reference}$ with no temperature change
and with climate conditions comparable to the sites of origin. We further applied an irrigation treatment (+12-
21 % of ambient precipitation) and an N-deposition treatment (+3 kg and +15 kg N $ha^{-1}$ $a^{-1}$), the latter simulating
a fertilizing air pollution effect.
Moderate warming led to increased productivity. Across the four-year experimental period, the mean annual
yield peaked at intermediate CSs (+43 % at +0.7 °C and +44 % at +1.8 °C), coinciding with c. 50 % of days
with dry soil during the growing season (growing-season-days with soil moisture <40 %). The yield increase
was smaller at the lowest, warmest CS (+3.0 °C), but was still 12 % larger than at $CS_{reference}$. Days with dry soil
explained the average yield-differences among CSs well. Irrigation had a significant effect on yield (+16-19 %)
in dry years, whereas atmospheric N-deposition did not result in a significant yield response. We conclude that
productivity of semi-natural, highly diverse subalpine grassland will increase in the near future. Despite
increasingly limiting soil water content, plant growth will respond positively to up to +1.8 °C warming during
the growing period, corresponding to +1.3 °C annual mean warming.



## 1 Introduction

The present period of global warming is most pronounced in the cold regions of high altitude and high latitude (Core writing team, IPCC 2014). The productivity of these ecosystems is temperature-limited, and even though the temporal distribution of total annual radiation differs, they share many similarities. After the temperature decline following the Holocene climate optimum (ca. 9000 - 6000 a BP; Vinther et al., 2009), they are now experiencing a rapid rewarming.

In cold environments, the perspective on climate change is different compared to temperate and warm environments. First, mitigation of the thermal growth limitation is likely to have beneficial effects. Second, the warming-associated drought-risk is lower. The evaporative demand is much lower and at least the initial water supply for plant growth is granted, because even a small winter snowpack supplies a large soil moisture resource in spring. Third, in many regions the warming comes along with rising atmospheric nitrogen (N) deposition, originating from agriculture and fossil fuel burning. Atmospheric N deposition can be as little as <5 kg N ha$^{-1}$ a$^{-1}$ at remote mountain sites (Rihm and Kurz, 2001), but can reach rates >40 kg N ha$^{-1}$ a$^{-1}$ elsewhere in Switzerland (Rihm and Achermann, 2016). This fertilizing air pollution agent promotes plant growth and has the potential to reduce plant species diversity by favoring fast growing species (Vitousek et al., 1997; Bobbink et al., 2010; Phoenix et al., 2012). Alone and in interaction, all three factors increase the ecosystem plant productivity potential.

However, previous warming experiments on plant productivity have shown inconsistent results. For example, tundra vegetation showed a twofold productivity increase, driven by increased summer temperature (Van der Wal and Stien, 2014). In contrast, Liu et al. (2018) combined long-term observations with a manipulative experiment to find that total net primary productivity (NPP) in Tibetan grassland remained unaffected, though grasses were favored over forbs and sedges by drought and warmth. In yet another meta-analysis, only 13 out of 20 experimental grassland sites revealed small increases of plant productivity due to warming (Rustad et al., 2001): while grassland ecosystems in general showed both positive and negative responses, the colder tundra systems (high latitude or altitude) with lower precipitation had positive and larger productivity responses to warming. Given that essential ecosystem services strongly co-depend on plant productivity (e.g., forage supply for livestock and wildlife, soil erosion control and support of the biological carbon sink), an improved knowledge on how climate warming affects productivity of colder grassland-systems is required.

A common restriction for the usability of climate change experiments for ecosystem productivity projections lies in the low number of concurrently manipulated environmental factors (Rustad 2008; but see Dukes et al., 2005 for an exception). This potentially leads to an overestimation of effects when data from several, single factor experiments are combined in meta-analyses or models (Leuzinger et al., 2011). Indeed, productivity responses to combined factors are usually less than additive in size, compared to single treatment responses (Dieleman et al., 2012; Xu et al., 2013). Not only can a low number of treatment factors, but also a low number of treatment levels invite overly simplistic interpretation of experimental results, if only a short or linear segment out of a larger range of biologically possible responses is represented in the data. For example, a hump-shaped response curve (2-dimensional) under atmospheric N-deposition best described the properties of a soil C-sink in subalpine grassland (Volk et al., 2016). Similarly, a ridge-shaped response surface (3-dimensional), driven by temperature and precipitation during 17 experimental years, was needed to explain NPP data (Zhu et





al., 2016). These findings suggest that the outcome of a global change productivity-experiment depends to a
large degree on the chosen factor levels and their interaction with the ambient climate during the experiment.
Here, we present four-years of treatment results from a field experiment in the Swiss Alps. We used a variety of
grassland communities by transplanting turf monoliths from six different sites of origin to one common
experimental site, to observe a plant productivity response that is not restricted to a specific species
composition. Turf monoliths were distributed over six levels of altitude to generate a climate gradient. Doing so,
we included not only the temperature change, but also the changing length of the growing period. The between-
year weather variability created a large variety of climate situations within the range of potential growth
conditions. Additionally, a two-level irrigation treatment and a three-level atmospheric N-deposition treatment
were set up. We hypothesized that

1) The effect of warming on plant growth would be beneficial at moderate warming levels, but detrimental at

high warming levels.

2) Increased soil water content would mitigate the detrimental effects of excessive warming levels.

3) N-deposition would exhibit a generally favorable effect on plant growth. This effect would further

increase with higher temperatures and irrigation due to their mitigating effect on thermal and water co-

limitations.





## 2 Materials and Methods

This experiment (AlpGrass experiment) used grassland monoliths (MLs) to investigate climate change effects on subalpine pasture ecosystems in the central Alps. At six different sites in the Canton Graubünden, Switzerland, areas of 1 ha on southerly exposed, moderate slopes were selected at an altitude of ca. 2150 m a.s.l. to serve as 'sites of origin'. All six sites were mountain grassland used for summer livestock grazing, within ≤ 55 km distance of each other, but their soil (typical depth 20-30 cm) developed either on basic or on acidic bedrock. Thus, the sites of origin shared very similar climatic conditions, but represented a wide range of soil properties and plant communities. Plant communities at the sites of origin were generally dominated by grass and sedge species, but comprised also a substantial share of forb and few legume species. Extensive information on soil properties and species composition of the different origins can be found in Wüst-Galley et al. (2020).

In June 2012 a total of 252 MLs (6 sites of origin × 42 MLs) of 0.1 m$^2$ surface area (L × W × H = 37 × 27 × 22 cm) were excavated at the sites of origin. Randomly generated X-Y-coordinates were used to choose the location of excavation. If a distinct location had sufficiently deep soil and no rocks, if bare soil and woody species were < 10 %, and if there was no apparent dominance of single plant species, then MLs were extracted. Else, the next pair of coordinates was probed. MLs were placed into precisely-fitting, well-drained plastic boxes to facilitate future transport and avoid potential side effects of experimental treatments applied later. To minimize the disturbance of temperature and moisture conditions, MLs were immediately reinserted into the ground at their respective site of origin.

Half a year later, in November 2012, 36 MLs were transported from each site of origin to the common AlpGrass experimental site, while 6 MLs each remained at their original site to allow for an assessment of the transplanting effect. Standardizing harvests were done in 2012 (zero-year) and 2013 (acclimation), while quantitative harvests used in this analysis continued from 2014 to 2017.

## 2.1 Experimental site and treatment design

The AlpGrass experimental site is located on the south slope of Piz Cotschen (3029 m), above Ardez in the Lower Engadine valley (Graubünden, Switzerland). The site as a whole covers a 680 m altitudinal gradient, characterized by a vegetation change from montane forest (WGS 84 N 46.77818°, E 10.17143°) to subalpine grassland (WGS 84 N 46.79858°, E 10.17843°). Along the gradient, six separate climate scenario sites (CS) were located at different altitudes (CS1: 2360 m, CS2: 2170 m, CS3: 2040 m, CS4: 1940 m, CS5: 1830 m, CS6: 1680 m a.s.l.). Because CS2 had a similar altitude as the sites of origin, it was chosen as a reference site (hereafter CS2$_{reference}$). CS2$_{reference}$ and sites of origin are all characterized by cold winters with permanent snow cover. The snow-free period lasts approximately from May to October, with a mean April – October air temperature of 6.5 °C during the experiment (Tab. 1). Annual mean temperature at CS2$_{reference}$ was 3.2 °C and mean precipitation sum was 748 mm (Tab. 2).

At each of the 6 CS, 6 MLs from each of the six sites of origin were installed in the ground within their drained plastic boxes, flush with the surrounding grassland surface, resulting in 36 MLs per CS and a total of 216 transplanted MLs. Monoliths were arranged side by side without a separating gap or buffer zone; occasional gaps between MLs and the surrounding turf were filled with soil to prevent air flow. The grassland surrounding the MLs was frequently mown to prevent the introduction of new species/genotypes by seed dispersal.





At each CS, an irrigation and an N-deposition treatment were set up in a cross-factorial design. One half of the
36 MLs (3 MLs per site of origin) received only ambient precipitation and no additional water, the other half
received additional water during the growing season. Within each irrigation treatment, MLs were subjected to an
N treatment representing three levels of atmospheric N-deposition (treatment details below, and see Appendix
Tab. A1 for a schematic description). At each CS, irrigation and N treatments were arranged in a randomized
complete block design (six blocks each containing all six irrigation × N treatment combinations). Moreover,
MLs of the six sites of origin were assigned to the six blocks by restricted randomization so that an equal
distribution of sites of origin to all blocks was ensured.

| Site | Alt. (m) | Air temp. (Mean, °C) ±1SE | | $\Delta T$ (°C) | $DD0°C_{total}$ | Pre-harvest period | |
|---|---|---|---|---|---|---|---|
| | | Apr. – Oct. | annual | Apr. – Oct. | Mean ±1SE | # Days | ±1SE |
| CS1 | 2360 | 5. ±0.17 | 1.6 ±0.20 | -1.4 | 1156 ±50 | 78 | ±4.3 |
| $CS2_{reference}$ | 2170 | 6. ±0.17 | 3.2 ±0.23 | 0.0 | 1440 ±43 | 91 | ±3.8 |
| CS3 | 2040 | 7. ±0.17 | 3.7 ±0.20 | 0.7 | 1649 ±67 | 107 | ±4.4 |
| CS4 | 1940 | 8. ±0.16 | 4.7 ±0.25 | 1.5 | 1746 ±71 | 104 | ±2.8 |
| CS5 | 1830 | 8. ±0.17 | 4.6 ±0.21 | 1.8 | 1829 ±10 | 97 | ±3.4 |
| CS6 | 1680 | 9. ±0.17 | 5.8 ±0.21 | 3.0 | 2095 ±14 | 104 | ±3.5 |

**Table 1** Climatic parameter means across years (±1SE) at the climate scenario sites (CS) during the experiment:
Mean air temperature from April to October and for the whole year, April – Oct. air temperature difference ($\Delta$
T) of respective CS' compared to $CS2_{reference}$. Degree days above 0 °C for the snow free period between annual
harvests ($DD0°C_{total}$). Pre-harvest period length is the number of days between snow-melt and harvest.


| Site | Alt. (m) | Precipitation (sum, mm) | | Dry days (%) | | Harvest |
|---|---|---|---|---|---|---|
| | | Apr. – Oct. | annual | not irrigated | irrigated | Date (Ø) |
| CS1 | 2360 | 674 ±18 | 752 ±20 | 27 ±5.3 | 17 ±5.1 | 12 Aug |
| $CS2_{reference}$ | 2170 | 656 ±27 | 748 ±27 | 31 ±1.7 | 20 ±2.7 | 26 July |
| CS3 | 2040 | 629 ±26 | 732 ±21 | 42 ±5.2 | 24 ±4.3 | 22 July |
| CS4 | 1940 | 614 ±20 | 739 ±22 | 33 ±2.2 | 24 ±3.5 | 14 July |
| CS5 | 1830 | 628 ±20 | 780 ±17 | 55 ±4.4 | 41 ±5.0 | 09 July |
| CS6 | 1680 | 570 ±19 | 687 ±21 | 73 ±3.1 | 53 ±4.5 | 05 July |

**Table 2** Precipitation sums for the climate scenario sites, aggregated from April to October and annually. For
comparison: The closest Swiss Federal Office for Meteorology station (Scuol, 1303 m a.s.l., 9 km distance)
reported 662 mm mean annual precipitation during the experiment. Dry days (%) indicates the percentage of
days during the pre-harvest period with SWC <40 %. The phenology triggered harvest date reflects the delayed
vegetation development at higher altitudes.





### 2.2 Climate scenario site (CS) climate change treatment

The different altitudes of the CSs created a climate change scenario treatment, commencing in November 2012,
when the MLs were installed at the AlpGrass site, and ending in 2017 with the final harvest. The difference in
altitude between the sites of origin and the respective CS at the AlpGrass experimental site determined the
change of climatic conditions that the transplanted MLs experienced. These conditions include the mean
growing period temperature, from April to October. We assumed the evenly moderate temperature (ca. 0 °C)
under the winter snow cover to be of little importance for differences in ecosystem productivity. The CS
temperature treatment was specified as the deviation from $CS2_{reference}$ temperature. The thermal energy was
expressed as degree day values (DD0°C), resulting from hourly air temperature means above a threshold of
0 °C, added for one day, then divided by 24. To quantify the total thermal energy available we summed degree
days during the snow-free period between the annual harvests ($DD0°C_{total}$), considering that the perennial
vegetation continues to grow after mowing.
Differences in volumetric soil water content (SWC) were quantified as 'percent (%) dry days'. This represents
the proportion of days during the growing period with a SWC < 40 %. The < 40 %-threshold does not
necessarily imply strong plant growth limitation, but it reliably provided a good contrast for differences in the
soil moisture status between the CSs and between years.

### 2.3 Irrigation treatment

An irrigation treatment with two levels was set up to distinguish the warming effect from the soil moisture
effect, driven by warming. In several applications throughout the growing period, precipitation equivalents of 20
mm were applied to the MLs under the irrigation treatment. The total amount of water added per ML was 80,
120, 120 and 80 mm in 2014, 2015, 2016 and 2017, respectively. These amounts were equivalent to 12-21 % of
the recorded precipitation sum during the growing periods.

### 2.4 N-deposition treatment

The N-deposition treatment consisted of three levels. Atmospheric N-deposition from air pollution was
simulated to amount to a deposition of 3 and 15 kg N ha$^{-1}$ a$^{-1}$, on top of the present background deposition. We
used a 200 ml ammonium nitrate ($NH_4^-$ $NO_3^-$)/water solution per monolith, which was applied in twelve, ca. bi-
weekly fractions, covering the growing period. Monoliths without additional N-deposition received water
without ammonium nitrate.

### 2.5 Meteorology

At all six CS we measured air temperature, relative humidity (Hygroclip 2 in an unaspirated radiation shield,
Rotronic, Switzerland), and precipitation (ARG100 tipping bucket raingauge, Campbell Scientific, UK). Soil
temperature and SWC were measured at 8 cm depth in 6 MLs each at topmost CS1 and intermediate CS3, CS4
and CS5, using a SWC reflectometer with 12 cm rods (CS655, Campbell Scientific, UK). At $CS2_{reference}$ and
lowest CS6 these values were measured in 18 MLs and two points in the surrounding grassland. The
measurement interval for all parameters was 10 minutes originally and was later integrated for longer periods as
necessary.





At each site of origin we installed Hobo U12-008 data loggers with TMC-HD sensors (Onset Computer
Corporation, USA) in three monoliths and one spot in the undisturbed, surrounding grassland for comparison
with the reference climate scenario site $CS2_{reference}$.
Ambient wet N-deposition was measured at $CS2_{reference}$ and lowest CS6 using bulk samplers (VDI 4320 Part 3,
2017; c.f. Thimonier et al., 2019) between April 2013 and April 2015. Nitrate ($NO_3^-$) in rainwater and melted
snow was analyzed by ion chromatography (ICS-1600, Dionex, USA) and $NH_4^+$ was analyzed using a flow
injection analyzer (FIAstar 5000, Foss, Denmark) with gas diffusion membrane, detection was completed with
UV/VIS photometry (SN EN ISO 11732).

**2.6 Plant productivity**
All plant material (including mosses and lichens) of the MLs was cut 2 cm above the soil surface once per year
at canopy maturity. This plant removal serves as a proxy for the short, but intensive summer grazing period of
the traditional management. As a result of the phenology-triggered harvests (anthesis of *Festuca rubra*), the
topmost CS1 was cut on average 38 days later than the lowest CS6. Plants were dried at 60 °C, allowed to cool
in a desiccator and weighed to determine dry matter yield (hereafter biomass yield).

**2.7 Data analyses**
Data were analyzed by linear mixed-effects regression. First, we were interested in the overall response of
biomass yield over years as affected by the treatment factors. To this aim, biomass yield was averaged across
the four experimental years (2014-2017) and was regressed on CS (factor of 6 levels), irrigation (factor of 2
levels), and N-deposition (factor of 3 levels), including all interactions. 'Site of origin' (6 sites) and block (36
levels: 6 CS × 6 blocks) were modelled as random factors (random intercepts). Restricted maximum likelihood
was used for parameter estimation. For the inference on fixed effects, the Kenward–Roger method was applied
to determine the approximate denominator degrees of freedom (Kenward and Roger 1997), and the marginal
and conditional $R^2$ of the model were computed following Nakagawa and Schielzeth (2013). Differences in
biomass yield between single CSs and the $CS_{reference}$ were tested based on the model contrasts (post-hoc *t*-tests,
without using multiple comparisons). To receive additional insight into within year treatment effects, this very
same model was also applied to data of each of the four individual years.
Second, to consider the time effect and the repeated structure of the data, biomass yield of all four years was
regressed on year (factor of 4 levels), CS, irrigation, and N-deposition (factor levels as described), including all
interactions. Here, random factors consisted of an identifier for MLs (216 levels) to consider the potential
correlation of MLs' biomass yield over years (modelled as random intercept). In addition, the model included
the random factor 'site of origin' and allowed for a separate block term at each of the four years (details as
described). Residuals of all models were evaluated for normality and homoscedasticity and fulfilled assumptions
of linear mixed-effects regression. Finally, to gain insight into effects of thermal energy and drought on plant
productivity, biomass yield was modeled as function of each $DD0°C_{total}$ and percent dry days using generalized
additive models (GAM). Generalized additive models had to be used as simple linear models could not
appropriately handle these relationships. The GAMs included the fixed factor irrigation and a smooth term for
the continuous variables $DD0°C_{total}$ and percent dry days, respectively, for both levels of the irrigation treatment.
Model validation revealed that the assumptions of GAMs were met. All data was analyzed with the statistics





software R, version 4.0.0 (R Core Team 2020) and packages lme4 for linear-mixed effect models (Bates et al.,
2015) and mgcv for GAMs (Wood, 2017).



## 3. Results

### 3.1 Climate scenario site (CS) environmental conditions

#### 3.1.1 Low atmospheric background N-deposition

Total N-deposition was 3.3 kg N ha$^{-1}$ a$^{-1}$ at CS2$_{reference}$ and 4.3 kg N ha$^{-1}$ a$^{-1}$ at the lowest CS6. The seasonal distribution showed peak deposition rates in June and July.

#### 3.1.2 Consistent temperature, precipitation and drought changes with altitude

The mean Apr. – Oct. temperature gradient of up to +3 °C compared to CS2$_{reference}$, distributed over four altitudinal levels (CS3 – CS6), constituted the warming treatment. Conversely, temperature at the topmost CS1 constituted a cooling treatment (Δ temp. -1.4 °C), extending the range of temperature responses tested (Δ temp., Tab. 1). As intended, the DD0°C$_{total}$ steadily increased from CS2$_{reference}$ to lowest CS6. The pre-harvest period (PHP) length was fairly similar among CSs, because the early snow-melt at the lower CS was compensated by an early harvest (Tab. 1).

We observed a small, non-continuous increase of precipitation with altitude during April – October. The recorded annual precipitation sum was somewhat larger than the sum for the growing period (Tab. 2). The length of the period with dry soil (% dry days) doubled along the altitudinal gradient: At the two top CSs only one third of the pre-harvest period was dry, compared to two thirds of the time at the lowest site CS6 (compare Tables 1 & 2). The irrigation treatment reduced the incidence of dry days to 60-80 % of the non-irrigated situation (Tab. 2).

#### 3.1.3 Small transplantation effects on soil temperature and moisture

At the sites of origin, the mean April – October soil temperatures in the undisturbed grassland were 8.8 ° (±0.3) compared to 8.9 °C (±0.3) in the monoliths. At CS2$_{reference}$ this difference was 9.2 ° vs. 9.5 °C. Thus, the surrounding grassland at CS2$_{reference}$ site was on average 0.4 °C warmer than at the sites of origin, and monoliths at CS2$_{reference}$ were 0.3 °C warmer than the undisturbed grassland surrounding the experiment. Volumetric SWC in the undisturbed grassland was 1 % lower on average compared to SWC in the monoliths at CS2$_{reference}$ and lowest CS6.

### 3.2. Yield

#### 3.2.1 Insignificant transplantation effect

The mean annual yield was 20 % larger at CS2$_{reference}$ (control treatment MLs), compared to the origins (162 g m$^{-2}$; ±12.7), but not significantly different ($P = 0.19$; paired, two-sided $t$-test). Equally important, the difference showed no trend, as in some years the yield at CS2$_{reference}$ was higher, in some years it was lower compared to the sites of origin.

#### 3.2.2 Strongest climate scenario site effect at intermediate CS

Across the four years, we found a highly significant effect of the CS on aboveground biomass yield (Tab. 3). Intermediate warming increased yields by +43 %, +18 % and +44 % at sites CS3, CS4 and CS5, respectively (Tab. 4, $P \leq 0.05$ at least). Even at the warmest site CS6 the yield was still +12 % larger compared to the



$CS2_{reference}$ site. The coldest site CS1 was not less productive than $CS2_{reference}$. In the year of the overall
maximum productivity (2016), both the coldest site CS1 and the warmest site CS6 produced their respective
record yield (Tab. 4). Overall, the yields of the 24 combinations of year × CS varied by a factor of 2.1 (yields
averaged across irrigation and N-deposition treatments). The yield response to CSs differed between years
(Appendix Tab. A2, year × CS interaction: $P < 0.001$) in that the CS effect became weaker towards the end of
the experiment (Appendix Tab. A3).


| Variable | $df_{num}$ | $df_{den}$ | $F$-value | $P$ |
|---|---|---|---|---|
| Climate Scenario (CS) | 5 | 29.1 | 14.9 | < 0.001 |
| Irrigation | 1 | 145.2 | 6.5 | 0.012 |
| N | 2 | 145.2 | 1.3 | 0.287 |
| CS × Irrigation | 5 | 145.2 | 1.1 | 0.352 |
| CS × N | 10 | 145.2 | 0.5 | 0.864 |
| Irrigation × N | 2 | 145.2 | 1.1 | 0.348 |
| CS × Irrigation× N | 10 | 145.2 | 1.3 | 0.241 |

df$_{num}$: degrees of freedom of term; df$_{den}$: degrees of freedom of error (which can be fractional

in restricted maximum likelihood analysis)

**Table 3** Summary of analyses for the effects of climate scenario site (CS), irrigation and N deposition on
aboveground biomass yield of subalpine grassland. Data were averaged across the four experimental years (total
n = 216). $F$-tests refer to the fixed effects of the linear mixed-effects model. The marginal and conditional $R^2$
were 0.41 and 0.50, respectively.




Aboveground biomass yield (g m$^{-2}$, means ±1SE)

| | CS2$_{reference}$ | | | CS1 | | CS2$_{reference}$ | | CS3 | | CS4 | | CS5 | | CS6 | | CS mean | |
|---|---|---|---|---|---|---|---|---|---|---|---|---|---|---|---|---|---|
| Year | % dry days | DD0°C$_{total}$ | | | | | | | | | | | | | | | |
| 2014 | 30 | 1353 | | 149$^{ns}$ | ±8.0 | 170 | ±11.0 | 238*** | ±8.8 | 203* | ±11.6 | 255*** | ±15.2 | 152$^{ns}$ | ±10.5 | 194 | ±5.3 |
| 2015 | 38 | 1359 | | 147$^{ns}$ | ±8.1 | 138 | ±5.8 | 248*** | ±12.1 | 171$^{†}$ | ±8.9 | 310*** | ±13.6 | 198*** | ±8.8 | 202 | ±5.7 |
| 2016 | 22 | 1509 | | 230$^{ns}$ | ±8.7 | 222 | ±9.1 | 297*** | ±10.2 | 247ns | ±11.1 | 271** | ±15.3 | 250$^{†}$ | ±9.8 | 253 | ±4.7 |
| 2017 | 34 | 1541 | | 152$^{ns}$ | ±8.5 | 166 | ±7.8 | 208* | ±10.0 | 201* | ±11.7 | 169$^{ns}$ | ±9.1 | 176$^{ns}$ | ±8.3 | 178 | ±4.0 |
| Mean | 36 | 1440 | | 170$^{ns}$ | ±7.1 | 174 | ±6.9 | 248*** | ±7.9 | 205* | ±9.0 | 251*** | ±11.5 | 194$^{ns}$ | ±6.9 | | |

*** $P \leq 0.001$, ** $P \leq 0.01$, * $P \leq 0.05$, † $P \leq 0.1$, ns $P > 0.1$

**Table 4** Aboveground biomass yield (means ±1SE) per CS and year, averaged across irrigation, N-deposition treatments, and site of origin. Within each year, significance tests are against CS2$_{reference}$, based on the model contrasts derived from linear mixed-effects regression (see Appendix Tab. A1, for the respective model summary).



### 3.2.3 Irrigation effect in dry years

Despite a mere +7.7 % average yield increase (Fig. 1 B), irrigation turned out to be a significant factor across years
(Table 3). Yet, the effect of irrigation differed between years (Appendix Tab. A2, year × irrigation interaction: $P <$
0.001), and single years analysis detected positive effects of irrigation only in 2015 (+15.8 %) and 2017 (+18.8 %)
(Appendix Tab. A3). In these years, the percentage of days with dry soil was highest.

### 3.2.4 No nitrogen deposition effect

Five years of experimentally increased atmospheric nitrogen deposition (+3 and +15 kg N ha$^{-1}$ a$^{-1}$) did not cause a
significant response of biomass yield (Fig. 1 A; Tab. 3). Moreover, there was no significant interaction detected
between the N-treatment and the factors CS or irrigation. Single years analysis, to test for a late response to
accumulating amounts of N, revealed a marginally significant effect only in 2016 (Appendix Tab. A3).

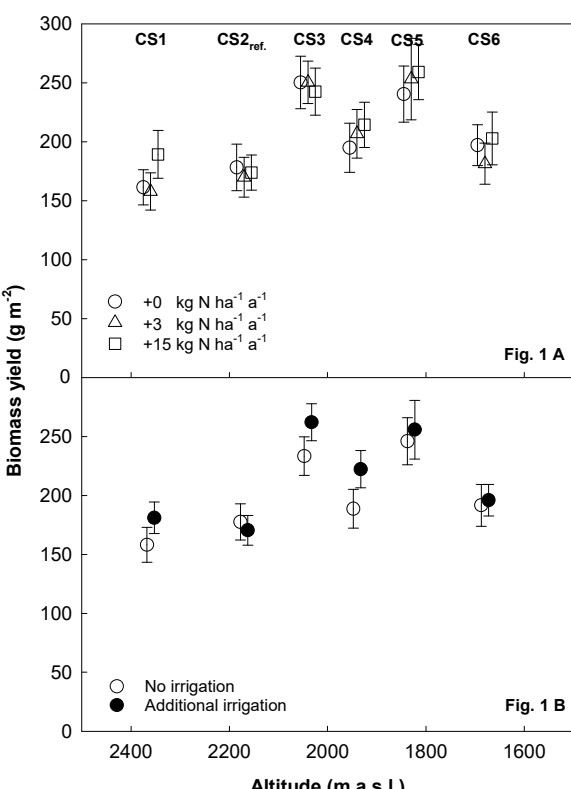

**Figure 1 A, B** Aboveground biomass yield as a function of the altitude of CSs. Data were averaged across years;

circles denote means ±1 SE. Warming and dry days (%) increase with decreasing altitude from left to right. **A)** Yield



values grouped by N-deposition treatment (0, 3 and 15 kg N ha$^{-1}$ a$^{-1}$, in addition to 4-5 kg N background deposition).
**B)** Yield values grouped by irrigation treatment. Overlapping means and SEs are shifted horizontally to improve
their visibility.

**3.2.5 Climate scenarios strongly relate to temperature and soil moisture changes**
Biomass yield at the different CSs was significantly related to thermal energy, expressed as growing DD0°C$_{total}$.
Here, intermediate CSs (CS3, CS5) had greatest yields at intermediate values of DD0°C$_{total}$, indicated by the
curvature of the predicted line in particular under irrigated conditions (Fig. 2 A, Appendix Tab. A4, smooth term for
DD0°C$_{total}$: $P < 0.001$).
Similarly, biomass yield was significantly related to soil moisture content, expressed as percent of days with dry soil
(SWC < 40 %) during the growing season, with intermediate CS3 and CS5 having highest yields at around 50 % of
dry days under no irrigation and at around 30 % dry days under additional irrigation (Fig. 2 B, Appendix Tab. A5,
smooth term for dry days: $P < 0.001$). Under unirrigated conditions, in parallel with a doubling of dry days (from
27 % at topmost CS1 to 55 % at intermediate CS5), yield consistently rose and only fell at the driest and warmest
site CS6, with 73 % dry days.

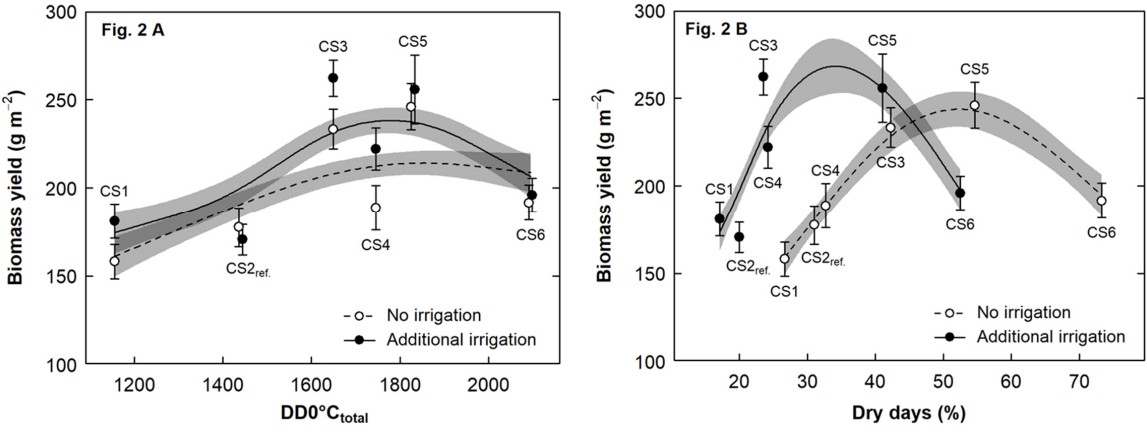

**Figure 2 A, B** Aboveground biomass yield at the six CS as **A)** a function of total received thermal energy
(DD0°C$_{total}$), and **B)** percent of days with dry soil (SWC < 40 %) during the growing season (dry days %). Data were
averaged across years; circles denote means ±1 SE per CS and irrigation treatment. The predicted line is based on a
generalized additive model using all data (±1 SE light grey shaded). Dark grey indicates the cross-section of the two
SE bands. Overlapping means and SEs in A) are shifted horizontally to improve their visibility.



## 4 Discussion

We found a substantial and significant positive effect of climate scenario warming (up to + 1.8 °C) on aboveground
biomass yield of subalpine grasslands (up to +44 %). Contrary to expectation, additional resource supply through
irrigation and N-deposition had only marginal (water) or no effects (N) on yield, respectively. Our transplanting
experiment proved to be efficient in assessing several linked climate change drivers in their effect on plant growth.

### 4.1 Climate scenario temperature effects

The phenology-triggered harvest opened the possibility to extend the growing period in cool years and shorten the
exposure to drought stress in warm years. Thus, beneficial thermal effects were maximized, while detrimental
drought effects were minimized. As a consequence, we displayed the yield over a continuous x-axis of degree days
between harvests ($DD0°C_{total}$, Fig. 2 A). This represents the available thermal resource, associated with a particular
yield, much better than mean temperatures of CS, or categorical values for CS1-CS6.

In cold environments, the warming is so important because the metabolic growth processes, which utilize the
assimilated energy, are strongly temperature dependent, much more so than the assimilation process *per se* (Körner
2003). In a meta-analysis of grassland responses to warming that included 32 sites, distinctly positive warming
effects on growth were found in the colder portion of those ecosystems (Rustad et al., 2001), very similar to
responses in the subalpine grassland of the current study. Interestingly, also the response size of our effects is in the
same range as that reported by Rustad et al., (2001).

Plant growth at the intermediate climate scenarios that represented a warming of 0.7 °C, 1.5 °C and 1.8 °C (Apr.-
Oct.) clearly benefitted from greater warmth. However, the increase of responses was somewhat inconsistent (CS4
ca. +18 %, CS3 and CS5 both > +40 %), matching only partly our first hypothesis. The erratic response of
intermediate CS4 is likely the result of an interaction of micro-topography effects on climate that were not detected
by our meteorological measurements, cockchafer (*Melolontha melolontha*) infestation, or the occurrence of mast
years in some species at that CS. In the extreme treatment at lowest CS6 (+3 °C Apr.-Oct., +2.4 °C annual mean) the
increasingly positive response to warming finally ceased, but yield was still somewhat larger than at $CS2_{reference}$. This
demonstrates that the increased thermal growth resource compensated for a radically reduced soil water resource
(compare Figure 2 A & B).

Despite substantial cooling at topmost CS1, coinciding with a temperature decline of -1.4 °C, the mean yields for
CS1 and $CS2_{reference}$ were very similar (Tab. 4). This is indicative of a plant community that is well cold-adapted.
Indeed, local historical records from the Swiss Federal Office for Meteorology (MeteoSwiss) show that only 100
years ago the local April-October mean air temperature was 1.4-1.5 °C lower than today (30 a running mean,
courtesy P. Calanca using MeteoSwiss data from Segl-Maria site at 1804 m a.s.l.). In effect, the cooling upward-
transplantation represented a climatic time travel of 100 years into the past. Also the dramatic temperature dynamics
during the 12,000 a of the present Holocene interglacial suggest that temperature adaptations, contained in modern
plant genotypes, may actually match not only today's climate conditions. From this perspective, the undiminished
productivity at topmost CS1 is not surprising. Instead, it illustrates that assumed 'control' temperatures in warming



experiments only represent the most recent point of an extremely dynamic climatic history, with respect to the
genetic memory of plants.

**4.2 Climate scenario soil moisture effects**

The differences in soil moisture content that resulted naturally from 24 different climatic situations (6 CS/altitude
levels × 4 years) created a hump-shaped response curve of yield over drought (Fig. 2 B). This does imply that, with
decreasing altitude and increasing warmth, productivity rose despite more dry days.
The importance of soil moisture for plant growth has been shown predominantly in much drier grasslands, e.g., in
warmer prairie (Xu et al., 2013) or cold alpine grassland (Wang et al., 2013), where release from drought stress
benefitted growth. For example, along a temperature and altitude gradient in semiarid Tibetan alpine grassland,
productivity increased with altitude due to reduced drought stress, but despite decreasing temperatures. Only after an
800 m rise in altitude, productivity eventually became smaller, and further reduced drought stress did not constitute
a further advantage on plant growth (Wang et al., 2016).
In our experiment, soil moisture values and its proxies integrate information on moisture *and* temperature. Thus, the
two-dimensional growth response curve along the altitudinal gradient, peaking at the least detrimental situation
between moisture limitation and thermal limitation (Fig. 2 B), is analogous to the three-dimensional response
surface found in the Jasper Ridge experiment (Zhu et al., 2016). Unfortunately, our experiment did not produce a
sufficient number of data points for a 3-D presentation. Based on these results, we infer that a joint evaluation of soil
moisture and temperature is mandatory to reliably assess warming effects of climate change on plant growth in the
subalpine environment.

**4.3 Irrigation treatment**

We had assumed that increased SWC would mitigate detrimental effects of excessive warming. Surprisingly
however, the overall irrigation effect on yield was not very substantial, despite large differences in the percentages
of dry days during the growing season (Table 2, Fig. 2 B). Moreover, the positive responses did not increase
consistently with warmth, but were strongest at the intermediate CS3 and CS4 (Fig. 1 B). Analyses of individual
years showed that the two significant responses of annual yield to irrigation coincided with the two driest years. This
evidence suggests that maximum mitigation of (low) temperature limitation requires simultaneous release of water
limitation, while at the same time the amount of water applied in our study was insufficient to compensate for
increased evapotranspiration at CS5 and the warmest site CS6.

**4.4 N-deposition treatment**

We hypothesized a generally positive effect of N-deposition on plant growth. Historically, the responsiveness of
(sub-) alpine vegetation to improved nutrient supply was considered to be restricted due to an overriding effect of
thermal energy limitation. Yet studies with very high rates of N application (40-100 kg N ha$^{-1}$ a$^{-1}$; Körner et al.,
1997; Heer and Körner 2002) showed substantial yield responses, also at alpine sites. Low N-dose responses of total
plant yield may require N-accumulation over years or a compound interest effect in plant biomass. For example,





only in the seventh treatment year a strong, +31 % total yield growth response to 5 kg N ha⁻¹ a⁻¹ was reported by
Volk et al., (2014) from subalpine grassland.
Single key species on the other hand showed immediate positive responses to realistic N-deposition rates (20-25 kg
N ha⁻¹ a⁻¹; Bowman et al., 2006; Bassin et al., 2007; Inauen et al., 2012). Similarly, a low dose experiment (5-30 kg
N ha⁻¹ a⁻¹) found no total aboveground biomass response, but a species composition change (Bowman et al., 2012),
indicating a growth benefit for some species at the expense of others. However, such single species responses may
be only transient: a strong *Carex* species response to as little as 5 kg N ha⁻¹ a⁻¹ was recently found to cease after five
years (Bassin et al., 2009 and 2013).
Indeed, *Carex spp.* can support a positive N-deposition growth response, but only until warming and drought create
a competitive advantage for grasses over sedges (Liu et al., 2018; Wüst-Galley et al., 2020). Thus, the latest studies
suggest that there is a positive N-deposition × warming interaction on the response of *Carex spp*.
In our experiment, we found no significant overall effect of N-deposition on yield after five years and only a
marginal effect in one year. We thus conclude that the cold-adapted, mature and low productivity grassland either
responds with a >5 year time lag, or that the N-deposition treatment was below the critical load for aboveground
biomass responses.

**4.5 Transplantation**
The turf monoliths at CS2$_{reference}$ were only slightly warmer and moister compared to the sites of origin, suggesting a
low transplantation impact (we have found no transplantation effect data from other experiments to compare with).
However, within the experimental site similar temperature increases between CS2$_{reference}$ and CS3 caused a much
larger productivity increase (+43 %). We reason that this incongruence can be explained by the difference in melt-
out time, which was on average only 3 days earlier at CS2$_{reference}$ (julian day 118) than at the sites of origin, but 21
days earlier at CS3 than at CS2$_{reference}$. We thus assume that the substantially earlier start of the growing season
caused the stronger growth response, despite a similar temperature change. This effect, induced by the
transplantation of the grassland MLs along the altitudinal gradient, demonstrates the importance of integrating
multiple drivers in climate change experiments to allow for a multi-factor driven plant response.
In our study, the effect of altitude on photosynthesis substrate limitation was considered negligible, compared to the
climate effects. The assimilation conditions of alpine plants have been the subject of investigation for decades. Since
the theoretical considerations of Gale (1972) and the field studies by Körner and Diemer (1987) and Körner et al.,
(1988), a predominant 'altitude-tolerance' of photosynthesis is widely accepted. Relevant environmental parameters
that change with altitude (temperature, $CO_2$ and $O_2$ partial pressure, vapor-pressure deficit and photosynthetic
photon flux density) have antagonistic effects on assimilation efficiency (see Wang et al., (2016) for a recent
discussion on the topic).



**5 Conclusions**

Despite dwindling soil water content, the subalpine grassland growth increased to up to +1.8 °C warming during the growing period (corresponding to +1.3 °C annual mean), compared to present temperatures. Even at the maximum warming (corresponding to +2.4 °C annual mean) the yield was larger than at the reference site. At the same time -1.4 °C cooling during the growing period (corresponding to -1.7 °C annual mean) did not reduce plant growth. This implies that subalpine grassland productivity has likely not increased during the past century warming, but, despite growing soil moisture deficits, will do so with continued warming in the near future.



**Author contribution**

MV and SB designed the experiment, MV, ALW and SB conducted field work. MV and MS analyzed the data. MV led the writing of the manuscript, with significant contribution from MS. All authors contributed critically to the drafts and gave final approval for publication.

**Data availability**

Data will be made available immediately to individuals upon request by the corresponding author. At a later point data will be deposited upon request at a publicly accessible repository, according to Swiss federal research institution guidelines.

**Competing interests**

The authors declare that they have no conflict of interest

**Acknowledgements**

We received essential financial support through the Federal Office for the Environment (contract No. 00.5100.PZ / R442-1499). The Federal Office for Meteorology (MeteoSwiss) is kindly acknowledged for providing access to meteorological data. We thank Pierluigi Calanca for handling these data. The Gemeinde Ardez and Alpmeister Claudio Franziscus generously allowed us to work on the Allmend. N-concentration analyses courtesy of Forschungsstelle für Umweltbeobachtung (FUB-AG, Rapperswil, Switzerland). We are grateful to Robin Giger for his untiring support in the field and the lab, and to the scientific site manager Andreas Gauer, who was in charge of the field sites.





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





**Appendix for**

**The rising productivity of alpine grassland under warming,**
**drought and N-deposition treatments**

Matthias Volk[1], Matthias Suter[2], Anne-Lena Wahl[1], Seraina Bassin[1,3]
[1]Climate and Agriculture, Agroscope, Reckenholzstrasse 191, 8046 Zurich, Switzerland
[2]Forage Production and Grassland Systems, Agroscope, Reckenholzstrasse 191, 8046 Zurich, Switzerland
[3]Pädagogische Hochschule Schaffhausen, Ebnatstrasse 80, 8200 Schaffhausen, Switzerland

Corresponding author: Matthias Volk (matthias.volk@agroscope.admin.ch)





## Appendix Tables

**Table A1** Schematic layout of monolith (ML) arrangement at each CS of the AlpGrass experimental site. At each CS, six MLs from each of six sites of origin were transplanted, resulting in 36 MLs. Two irrigation and three N-deposition treatments were set up in a cross-factorial design, resulting in six irrigation × N treatment combinations, which were assigned to each of the six MLs per site of origin. The six irrigation × N treatment combinations were arranged in a randomized complete block design of six blocks. Regarding sites of origin, the MLs were assigned to the six blocks in a restricted randomization, so that an equal distribution of sites of origin to all blocks was ensured. It follows that the six MLs from each site of origin received all irrigation × N treatment combinations and were evenly distributed on the site. Displayed is a possible randomization of irrigation and N treatments per block; at each CS separate randomizations were calculated.

| Block 1 | | | Block 2 | | | Block 3 | | |
|---|---|---|---|---|---|---|---|---|
| W0.N15 | W1.N0 | W0.N3 | W0.N3 | W0.N0 | W1.N15 | W1.N15 | W0.N0 | W1.N0 |
| W0.N0 | W1.N3 | W1.N15 | W1.N3 | W0.N15 | W1.N0 | W1.N3 | W0.N15 | W0.N3 |

| | | | | | | | | |
|---|---|---|---|---|---|---|---|---|
| W1.N15 | W1.N0 | W0.N0 | W1.N3 | W0.N3 | W1.N0 | W0.N0 | W1.N0 | W0.N15 |
| W0.N3 | W0.N15 | W1.N3 | W0.N0 | W1.N15 | W0.N15 | W1.N3 | W1.N15 | W0.N3 |
| Block 4 | | | Block 5 | | | Block 6 | | |

W0: no additional water (ambient precipitation only), W1: additional water during growing period; N0: no N fertilizer, N3: 3 kg N ha$^{-1}$ a$^{-1}$, N15: 15 kg N ha$^{-1}$ a$^{-1}$





**Table A2** Summary of analyses for the effects of climate scenario (CS), irrigation, and N deposition on
aboveground biomass yield of subalpine grassland over four experimental years. *F*-tests refer to the fixed
effects of a linear mixed-effects model that included all four years for a repeated measures analysis. The
marginal and conditional $R^2$ were 0.68 and 0.80, respectively.

| Variable | $df_{num}$ | $df_{den}$ | *F*-value | *P* |
|---|---|---|---|---|
| Year | 3 | 45.5 | 66.2 | < 0.001 |
| Climate Scenario (CS) | 5 | 198.0 | 18.3 | < 0.001 |
| Irrigation | 1 | 166.6 | 6.2 | 0.014 |
| N | 2 | 166.6 | 1.2 | 0.304 |
| Year × CS | 15 | 63.0 | 9.6 | < 0.001 |
| Year × Irrigation | 3 | 450.5 | 13.6 | < 0.001 |
| Year × N | 6 | 450.5 | 0.9 | 0.492 |
| CS × Irrigation | 5 | 166.6 | 1.1 | 0.380 |
| CS × N | 10 | 166.6 | 0.5 | 0.882 |
| Irrigation × N | 2 | 166.6 | 1.0 | 0.365 |
| Year × CS × Irrigation | 15 | 450.5 | 2.9 | < 0.001 |
| Year × CS × N | 30 | 450.5 | 0.8 | 0.749 |
| Year × Irrigation × N | 6 | 450.5 | 1.4 | 0.199 |
| CS × Irrigation × N | 10 | 166.6 | 1.2 | 0.275 |
| Year × CS × Irrigation × N | 30 | 450.5 | 1.4 | 0.066 |

$df_{num}$: degrees of freedom of term; $df_{den}$: degrees of freedom of error (which can be fractional
in restricted maximum likelihood analysis)



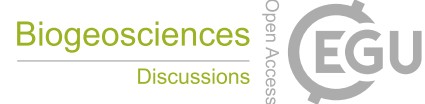

**Table A3** Summary of analyses for the effects of climate scenario (CS), irrigation, and N deposition on aboveground biomass yield of subalpine grassland at each of four experimental years (2014 – 2017). $F$-tests refer to the fixed effects of a linear mixed-effects model to each of the four years.

| Variable | $df_{num}$ | 2014 | | | 2015 | | | 2016 | | | 2017 | | |
|---|---|---|---|---|---|---|---|---|---|---|---|---|---|
| | | $df_{den}$ | $F$-value | $P$ | $df_{den}$ | $F$-value | $P$ | $df_{den}$ | $F$-value | $P$ | $df_{den}$ | $F$-value | $P$ |
| Climate Scenario (CS) | 5 | 28.9 | 17.2 | <0.001 | 29.5 | 24.9 | <0.001 | 29.3 | 4.5 | 0.004 | 29.4 | 4.0 | 0.006 |
| Irrigation | 1 | 145.2 | 1.5 | 0.224 | 145.1 | 21.6 | <0.001 | 145.3 | 1.1 | 0.290 | 145.4 | 19.2 | <0.001 |
| N | 2 | 145.2 | 0.7 | 0.481 | 145.1 | 0.5 | 0.610 | 145.3 | 2.6 | 0.078 | 145.4 | 0.3 | 0.728 |
| CS × Irrigation | 5 | 145.2 | 2.3 | 0.048 | 145.1 | 2.0 | 0.080 | 145.3 | 1.8 | 0.126 | 145.4 | 0.8 | 0.563 |
| CS × N | 10 | 145.2 | 0.5 | 0.912 | 145.1 | 0.7 | 0.751 | 145.3 | 0.9 | 0.531 | 145.4 | 0.5 | 0.896 |
| Irrigation × N | 2 | 145.2 | 1.9 | 0.151 | 145.1 | 0.8 | 0.448 | 145.3 | 0.7 | 0.509 | 145.4 | 1.2 | 0.290 |
| CS × Irrigation× N | 10 | 145.2 | 1.5 | 0.157 | 145.1 | 1.0 | 0.429 | 145.3 | 1.5 | 0.157 | 145.4 | 1.3 | 0.226 |

$df_{num}$: degrees of freedom of term; $df_{den}$: degrees of freedom of error (which can be fractional in restricted maximum likelihood analysis)



**Table A4** Summary of analyses for the effects of total received thermal energy (DD0°C$_{total}$) on aboveground
biomass yield of subalpine grassland under two irrigation treatments. Data were averaged across the four
experimental years (n = 216). *F*-values and approximate *P*-values refer to a generalized additive model that used
a smooth term for each irrigation treatment.

| Parametric terms | df | *F*-value | *P* |
|---|---|---|---|
| Irrigation | 2 | 1613.0 | < 0.001 |
| Smooth terms | edf | *F*-value | *P* |
| s(DD0°C$_{total}$) – No irrigation | 1.72 | 7.7 | < 0.001 |
| s(DD0°C$_{total}$) – Additional irrigation | 2.34 | 10.2 | < 0.001 |

df: degrees of freedom; edf: effective degrees of freedom (which can be fractional
in smooth terms of generalized additive models)
s: smoothing function applied on term

**Table A5** Summary of analyses for the effects of percent dry days during the growing season (dry days %) on
aboveground biomass yield of subalpine grassland under two irrigation treatments. Data were averaged across the
four experimental years (n = 216). *F*-values and approximate *P*-values refer to a generalized additive model that
used a smooth term for each irrigation treatment.

| Parametric terms | df | *F*-value | *P* |
|---|---|---|---|
| Irrigation | 2 | 412.6 | < 0.001 |
| Smooth terms | edf | *F*-value | *P* |
| s(Dry days %) – No irrigation | 2.55 | 11.9 | < 0.001 |
| s(Dry days %) – Additional irrigation | 2.70 | 10.7 | < 0.001 |

df: degrees of freedom; edf: effective degrees of freedom (which can be fractional
in smooth terms of generalized additive models)
s: smoothing function applied on term