# Peer review of "Sub-alpine grassland productivity increased with warmer and drier conditions, but not with higher N-deposition, in an altitudinal transplantation experiment"

_Biogeosciences, 2020_

## Referee Comment (RC1) · Anonymous Referee #1 · 13 Oct 2020

**1   General comments**

In this manuscript, the authors present a study crossing transplantation of subalpine grassland turfs from different elevations into a common garden with N fertilization and watering treatments. Because the average temperature at each turf's site of origin was different, they were subjected to different levels of warming above ambient by being placed in the same location. After four years, they found that productivity increased the most in turfs taken from intermediate elevations, and that watering increased productivity but fertilization did not. Overall the results are presented lucidly but I have a few concerns to address about the experimental design and the inference resulting from it, as well as some more minor comments below.

First, I would like to see more information about the plant species composition of the experimental monoliths. Qualitative results can be informative too. This could be a few sentences in the methods. Photographs might also be helpful.

Generally I would be slightly concerned about the inference obtainable from the elevational gradient. If many factors change in a correlated fashion along the gradient, such as temperature, moisture, and historical human/grazing pressure, it is hard to tell which factor is the driver. I understand the limitations of the design and I don't think it's necessarily a flaw, but this is something that should be addressed more openly. This is also the case when discussing how soil moisture integrates information on both temperature and moisture; this could also be viewed as confounding the effects of temperature and moisture.

One other point I would like to raise about the inference is that the warming treatment is confounded with site of origin. For example, the communities subjected to highest warming were those that were moved from the highest elevation. Therefore it is difficult to say whether the different levels of warming, or the composition of plant and soil communities from each of the sites of origin, led to the different productivity responses. This should be addressed as well.

For reproducibility, please make the code and data available in a repository so that readers can reproduce the results of the statistical analysis. This is especially important for the mixed model specification. Sometimes it is difficult for the reader to determine the exact model specification from the verbal description but it is easier if they can see the code.

**2   Line-by-line comments**

Line 10: The abstract does a good job of stating the results of the study but it does not do a good job of stating the motivation, novelty, or broader significance of the study

from the outset. Please revise accordingly.

Line 55: The claim that multifactorial experiments necessarily will improve predictions is debatable. Please expand on the reasoning behind this claim.

Line 76: The hypotheses need to have a little more justification or explicit statement of the reasoning why the particular directions of the effects and interactions are expected. For example, are there other studies that show similar effects or are the expectations derived from first principles?

Line 85: It is interesting that southerly exposed slopes were chosen for the study. They tend to be drier and warmer than slopes with different aspect at the same elevation. I would expect the plants living in these microclimates to be especially responsive to the warming treatment. Is this something worth briefly mentioning?

Line 106: A picture says 1000 words. It would be great to have some photos of the environment at the study sites, either as a main-text figure or as a supplement.

Line 116: Similar to above, it would be nice to have a picture of the experimental setup.

Line 150: Is there a justification for the threshold for growing degree days being set at 0C? The same goes for the 40% soil volumetric water content threshold.

Line 196 (statistical analysis section): I'm not sure I understand the reasoning behind assigning CS as a fixed effect but site of origin as a random effect. From my reading of the methods those are the same thing. Can you please clarify this?

Line 216: Please include some details on the GAM fitting procedure, such as functional form of splines, etc. Were the defaults from the mgcv package used? If important inference is drawn from the GAMs, it would be good to assess the sensitivity of the results to choices made in the GAM fitting process. As written, it is not reproducible.

Line 277: Because all columns of Table 4, besides the two leftmost, are in the same units (mean and SE of aboveground biomass yield), it might be better to convey the

information in this table with a figure. Currently it is difficult to visually extract the most salient patterns from the table. If you do not want to use a figure maybe another possibility would be to use colors or cell fills to show where the highest values in each year were recorded.

Line 281: I am confused why -7.7% is described as an increase, is it a negative or positive change?

Line 289: Refer to the statistical test result (I am assuming this is Table A2?) that supports the statement that there was no significant interaction between N treatment and CS or irrigation.

Line 316: "climate scenario warming" is a confusing phrase. Do you mean warming consistent with some particular climate scenario?

Line 390: I found this paragraph to be a little confusing. Are you referring to results from the present study or previous studies in the literature? Also, because you mention specific species responses to N addition from other species, it would be more interesting if you would draw a more direct connection with the present study. Were there any individual species that you can point to their responses?

Line 425: I am not sure what the grounds are for stating that subalpine grassland productivity will increase with warming. Is it necessarily the case that climatic conditions will "move up" in elevation – maybe there will be novel and unpredictable combinations of temperature and moisture not tested here.

---

## Referee Comment (RC2) · Anonymous Referee #1 · 13 Oct 2020

Thanks for the response! I am sorry that I misunderstood the methods description. Looking back it should have been obvious to me.

---

## Author Comment (AC1) · 13 Oct 2020

We are happy to have received the first referees' comments and are getting ready for a point by point reply as soon as the second review is available. There is one fundamental issue though, that we would like to address immediately:

Different from the referees interpretation, we do not create a warming treatment by transplanting turf monoliths from origins of different temperature to a common site of uniform temperature ('Because the average temperature at each turf's site of origin was different, they were subjected to different levels of warming above ambient by being placed in the same location.').

Instead, all sites of origin have very similar temperatures. But the experimental site,

where the turf monoliths are transplanted, contains a c. 700m altitudinal gradient. Along this gradient, six separate climate scenario sites are located at different altitudes. Thus, six different climate treatments are established.

We will pay maximum attention to avoid this misunderstanding in the future.

Kind regards, Matthias Volk

---

## Short Comment (SC1) · 14 Oct 2020

Never mind :-) ... it is always the authors task to be clear.

---

## Referee Comment (RC3) · Anonymous Referee #2 · 20 Oct 2020

Biogeosciences bg-2020-322: The rising productivity of alpine grassland under warming, drought, and N-deposition treatments

General Comments

In their manuscript titled "The rising productivity of alpine grassland under warming, drought, and, N-deposition treatments", the others describe a novel experiment in which monoliths of soil and turf were transplanted across an elevational gradient combine with fertilization and water addition treatments. After four years of growth in the transplanted location, the others describe how plant productivity in the monoliths responded to the interaction of different temperatures (comparing climate at the transplant location to the original site where the turfs were harvested from), fertilization, and increased moisture, as well as the interactive effects of these three treatments.
The results of this study showed that intermediate levels of warming increased plant productivity, even in drier conditions. Increasing the precipitation received by some monoliths had only marginal effects on plant productivity, while fertilizing the plots with nitrogen solutions had no discernable effect on plant productivity.

While this experiment is truly novel in its use of monolith transplants to simulate climate change in conjunction with two additional global change treatments in order to understand how multiple facets of global change will impact productivity, I have several concerns regarding the framing of these treatments, the metrics used to communicate and aggregate results, and the overall clarity of the manuscript. In particular, while transplanting monoliths to new elevations does of course impact climate, and in some cases results in warming, characterizing this experiment as a "warming experiment" is disingenuous. I encourage the authors to refer to their experiment as is, a transplant experiment across an elevational gradient. Furthermore, it is also a misnomer to refer to the precipitation manipulation component of this experiment as a "drought treatment", as water was added to some monoliths instead of removing precipitation, as when using rain-out shelters etc., to simulate drought. My detailed line comments below elaborate on these concerns as well as my suggestions and critique of the metrics that the authors chose to describe climate in this study.

Line Comments

34–"... to have beneficial effects": Beneficial effects on what?

35-36: Clarify what you mean by "initial water supply"... Water resources at the beginning of the growing season are generally plentiful? But this would be the case only for plants that emerge early in the growing season, i.e. depends on phenology of plant species.

38–"kg N ha-1 a-1": These units are unconventional, instead of a-1 (per annum?) I typically see yr-1 when describing nitrogen deposition rates.

45–"...showed a twofold productivity increase": In response to what treatment?

47–"...grasses were favored over forbs and sedges by drought and warmth": This seems unclear, what do you mean by "favored by drought and warmth"? Productivity of forbs and sedges increases with warming and drought?

61–"...if only a short or linear segment out of a larger range of biologically possible responses is represented in the data.": There is some indication that productivity relationships revealed in manipulative experiments actually encompasses even more variation than occurs naturally (see Jochum et al. 2020. Nature Ecology and Evolution).

67–I think that I am still confused by what you mean by "factor levels"... Does this refer to consideration of multiple global change factors, or does it refer to the magnitude of the global change treatment imposed by the experiments?

68– "Here, we present four-years of treatment results from a field experiment in the Swiss Alps.": This statement is an important introduction of your experiment, and as such, you should be more descriptive than "treatment results from a field experiment". What types of treatments specifically were involved in your field experiment, and were any of these treatments applied simultaneously to study interactive effects?

83–"monoliths (ML)": I do not feel that it is necessary to use an acronym for one word, and stating monolith regularly instead of ML will improve the clarity of your manuscript.

102-103: This sentence is rather unclear. What do you mean by standardizing harvests and the "zero-year" and "acclimation" distinctions? This aspect of your methods deserves an elaboration.

111-115: I find your naming convention, using the 'CS' designations, to be needlessly confusing. These are simply sites along an elevational gradient, so why not refer to them either by their numeric elevation (i.e. 2360 m) or simply as Elevation 1 (lowest elevation), Elevation 2.... etc., instead of introducing a less intuitive naming system.

116–"...6 CS, 6 MLs from each of the six sites of origin": I find your naming convention,

using the 'CS' designations, to be needlessly confusing. These are simply sites along an elevational gradient, so why not refer to them either by their numeric elevation (i.e. 2360 m) or simply as Elevation 1, Elevation 2.... etc., instead of introducing a less intuitive naming system.

119–"...were filled with soil to prevent air flow": Where did this soil come from? Bulk soil from each specific elevation/origin location?

121–"cross-factorial design": Full-factorial design? I'm unfamiliar with "cross-factorial" experimental designs.

153: This sentence is rather unclear... Temperatures were summed across one day?

154-156: This threshold seems particularly arbitrary, and I think that the use of a threshold in general is not necessary here. Why not simply present the mean growing season soil volumetric water content for each site/each season? This metric is much simpler and more intuitive for readers to understand and compare your results across the elevational gradient.

161-162: Why does the amount of precipitation added to each monolith vary between years?

168: Listing the chemical formula of ammonium nitrate is not necessary.

226: Is there some type of relationship between atmospheric N-deposition rates and elevation? Perhaps describe N-deposition rates across the entire gradient, not just at the middle and low points of your elevational gradient.

236: What does non-continuous mean? Non-linear?

239–"...only one third of the pre-harvest period was dry": It is definitely a misnomer to describe conditions of lower than 40% moisture content as "dry". In fact, in most alpine systems, 30% moisture content is considered ideal moisture conditions for optimal microbial activity (see Hawkes et al. 2017 PNAS for a relevant discussion related

to respiration and soil moisture). I would highly suggest re-characterizing the way in which you describe soil moisture in this manuscript. Instead of creating a binomial in soil moisture conditions around an arbitrary 40% moisture content threshold, why not just describe average soil moisture across the growing season on a continuous scale, i.e. just state average growing season soil moisture for the pre-harvest period.

248-249: Because you describe soil moisture conditions in the previous section using percent dry days, we have no way of understanding how this transplantation effect on soil moisture conditions (described using VWC) might interact with your other treatments.

251: I would suggest that productivity is the more appropriate term, consistent with literature in this area of ecological research, to describe your response variable.

259: In order to show evidence to support this claim, I would like to see a figure and the related statistics that shows the relationship between the productivity effect size (productivity in transplanted monoliths - productivity in control monoliths that were re-installed at the same site / standard deviation of productivity across all monoliths) regressed against the temperature difference from the monolith's original climate and the transplanted climate. In other words, how much of the change in productivity is explained by change in temperature following transplantation?

260-261: What does "intermediate warming" mean here? Describing this result as "monoliths that experienced X-Y degrees of warming by being transplanted to warmer climates at lower elevations relative to climate at their original location showed increases in productivity".

262-264: This sentence is confusing. 2016 was the year in which productivity, on average, was highest, but this was only the case at two sites? These two statements seem to contradict one another.

298: The title of this section seems to not relate to the results described within the

section. You already stated that each elevational site is characterized by different temperature and precipitation regimes in your methods and in previous sections of the results. Should this section describe the relationship between productivity and climate at each elevation?

325-326: Are there examples of other papers whose conclusions about the use of degree days instead of mean temperatures over the same time frame?

333-341: This section would benefit from a description of why the authors suspect that warming beyond "intermediate warming" was not associated with the same boost in productivity that was associated with intermediate warming.

337–"cockchafer (Melolontha melolonth) infestation: Please describe what this organism is and how it is relevant to variability in productivity.

347-349: Grammatical errors and diction in this sentence make it unclear.

358: I think this statement describes my point about eliminating your use of the "percent dry days" metric entirely... Your results, using this metric, prevent readers from relating the soil moisture conditions present in your experiment to soil moisture conditions elsewhere. Furthermore, describing soil moisture conditions less than 40% as "dry" is a misnomer.

380: What caused increased evapotranspiration at CS5? Is it possible that too much rainfall, either ambient or added as part of your irrigation treatment, could cause leaching of important soil nutrients, with higher VWC leading to lower productivity? This might be especially relevant in monoliths that received both an irrigation and fertilization treatment.

399-402: These are the only lines of this section of your discussion that reference your results directly. These sentences should be moved up in this section, and you should eliminate the references to other experiments with results that contradict what your experiment found, as this section is very unclear as currently written. Which of these

citations and theories help explain your results? Remove the rest.

426–"This implies that subalpine grassland productivity has likely not increased during the past century warming": This statement is in no way supported by your results.

---

## Author Comment (AC3) · 17 Nov 2020

> We are happy to read that the MS was received intelligible in general and we supply a comprehensive response to the issues that both reviewers identified. The reviewers' comments will help to substantially improve the MS. Below, please find the authors point by point replies. For ease of reading we quote the comments first, beginning with a '?' and start our responses with '>'

? First, I would like to see more information about the plant species composition of the experimental monoliths. Qualitative results can be informative too. This could be a few sentences in the methods. Photographs might also be helpful.

[Figure]

> We added more information on species directly in the new M&M. But to keep the MS as lean as possible we refer the specialist to the Wüst-Galley et al. (2020) paper on functional group responses in the same experiment and do not add photographs.

? Generally I would be slightly concerned about the inference obtainable from the elevational gradient. If many factors change in a correlated fashion along the gradient, such as temperature, moisture, and historical human/grazing pressure, it is hard to tell which factor is the driver.

> Indeed, in a complex system with a high number of interacting factors, it is usually not possible to point out one driver. Here, the soil moisture and temperature, resulting from the Climate Scenario (CS) site at a specific altitude, both drive the plant productivity response (cf. Fig. 2 A/B), but the fertilizing N deposition does not (cf. P values in Tab. 3). The management history of the sites of origin is very similar, but in concert with the strong edaphic factors an effect on the present plant communities cannot be excluded. We regard this element of heterogeneity as an advantage, as it is a factor that supports the general applicability of our results.

? I understand the limitations of the design and I don't think it's necessarily a flaw, but this is something that should be addressed more openly. This is also the case when discussing how soil moisture integrates information on both temperature and moisture; this could also be viewed as confounding the effects of temperature and moisture.

> True, the moisture of a Climate Scenario is not independent from the temperature. For this reason we termed the sites at different altitude 'Climate Scenario' and analyzed the data accordingly. We assumed that the unavoidable temp $\times$ moisture interaction closely resembles true climate change conditions, rather than an experimental manipulation of temp or moisture alone. To create a situation where this confounding is resolved, we introduced the irrigation treatment (P = 0.012; Tab. 3), that generated temperature independent moisture conditions. This concept is mentioned in chapter '2.3 Irrigation treatment'.

? One other point I would like to raise about the inference is that the warming treatment is confounded with site of origin. For example, the communities subjected to highest warming were those that were moved from the highest elevation. Therefore it is difficult to say whether the different levels of warming, or the composition of plant and soil communities from each of the sites of origin, led to the different productivity responses. This should be addressed as well.

> Different from the referee's interpretation, we do not create a warming treatment by transplanting turf monoliths from origins of different temperature to a common site of uniform temperature. Instead, all sites of origin have very similar temperatures and altitudes. But the experimental site, where the turf monoliths are transplanted, contains 6 climate scenario sites along a c. 700m altitudinal gradient. Thus, six different climate treatments are established. This misunderstanding was already clarified in an earlier round of comment/response. We will pay maximum attention to avoid this misunderstanding in the future.

? For reproducibility, please make the code and data available in a repository so that readers can reproduce the results of the statistical analysis. This is especially important for the mixed model specification. Sometimes it is difficult for the reader to determine the exact model specification from the verbal description but it is easier if they can see the code.

> This is a difficult issue. We respond to the specific statistics issue of the 'gamma' statement below at the comment on l. 216. If there are specific credibility issues, or data and code were to be used for further analysis, we would be happy to provide the necessary information at the desired level of integration. But we believe that the code and the carefully prepared dataset to go with the code does not prove anything. Alternatively, the raw data from the field are incomprehensible without a very, very extensive manual. Ultimately, even with all the data at hand, an (overall) 6 year, landscape-scale, climate change field experiment is not reproducible.

? 2 Line-by-line comments ? Line 10: The abstract does a good job of stating the results of the study but it does not do a good job of stating the motivation, novelty, or broader significance of the study from the outset. Please revise accordingly. > Helpful point. For the sake of brevity, we had opted for elaborating motivation, novelty and broader significance of the trial in the Introduction section. But we will add two lines in the Abstract that, together with the headline of the manuscript, will give the expert readers the necessary information to decide whether the article is of interest for them.

? Line 55: The claim that multifactorial experiments necessarily will improve predictions is debatable. Please expand on the reasoning behind this claim.

> The paragraph following our claim on l. 55 (l. 55-67) is dealing exclusively with the interpretation of multifactorial (or multilevel) vs. unifactorial experiments. It contains seven references to support the argument.

? Line 76: The hypotheses need to have a little more justification or explicit statement of the reasoning why the particular directions of the effects and interactions are expected. For example, are there other studies that show similar effects or are the expectations derived from first principles?

> Indeed, our hypotheses must be understood in the context of the Introduction, but the Introduction is not reiterated in the hypotheses. We aimed at making the Hypotheses the 'Conclusion' of the Introduction. For example, we hypothesize that '1) The effect of warming on plant growth would be beneficial at moderate warming levels, but detrimental at high warming levels.' (l. 76-77). As an introduction to the issue, the Introduction section mentions how warming at high altitude vs. lowlands reduces temperature growth limitations, rather than causing heat stress. Studies on the warming effect in cold environments (in part with inconsistent results) are discussed in detail (l. 44-54).

? Line 85: It is interesting that southerly exposed slopes were chosen for the study. They tend to be drier and warmer than slopes with different aspect at the same eleva-

tion. I would expect the plants living in these microclimates to be especially responsive to the warming treatment. Is this something worth briefly mentioning?

> True, southern exposure is warmer and drier. A uniformly southern exposure (and identical altitude) as opposed to different exposures was chosen to minimize differences in climate (temperature, moisture, radiation) between sites of origin. Also, in this region of the Alps the majority of southerly exposed slopes is used as summer pastures. In contrast, the more the slopes are pointing away from the sun, the more likely they are forested and not suitable for our grassland research. Surely one may assume that an adaptation to heat and drought has occurred in these plant communities, resulting in an improved tolerance against extreme events. This may make them more likely to outcompete plant communities from moist and cool habitats, as those grow warmer and drier. We do not see though, why plants from warm and dry habitats should be more responsive to warming, judged on a productivity increase/warming basis?

? Line 106: A picture says 1000 words. It would be great to have some photos of the environment at the study sites, either as a main-text figure or as a supplement.

> That is a hint we were waiting for ïĄŁ We would honestly love to show the site and the landscape it is set in. We will immediately start exploring the options. In a paperless publication environment, this should not be impossible!

? Line 116: Similar to above, it would be nice to have a picture of the experimental setup.

> Cf. above

? Line 150: Is there a justification for the threshold for growing degree days being set at 0C? The same goes for the 40% soil volumetric water content threshold.

> Indeed, the many degree day baselines for crops or other individual species are usually higher. They are empirically established to describe plant development stages that are important for plant cultivation or study. We chose a 'generic' 0°C baseline, because

in a mountain environment already low amounts of thermal energy play an important role. Also, lacking a single target species to focus on, but working on a multi-species community instead, none of these 'personalized' baselines appeared applicable. The SWC < 40 %-threshold does not imply plant growth limitation. Instead, it is an empirically developed contrast for differences in the soil moisture status between the CSs and between years. More time below the threshold simply means a drier period drier than less time below the threshold. This is also described in l. 158-161.

? Line 196 (statistical analysis section): I'm not sure I understand the reasoning behind assigning CS as a fixed effect but site of origin as a random effect. From my reading of the methods those are the same thing. Can you please clarify this?

> As clarified above, CS represents the climate scenario treatment along the altitudinal gradient at the AlpGrass experimental site. Because we wish to make specific tests and statements of the effect of the CS (or even one specific CS), this treatment has to be specified as a fixed effect, similar to the irrigation and the N-deposition treatments. By contrast, 'site of origin' represents the grasslands of the region where the monoliths were excavated and later transplanted to the AlpGrass site. We are not interested in and do not make any tests and statements on the effect of the grasslands of origin, therefore this variable should be fitted as a random factor.

? Line 216: Please include some details on the GAM fitting procedure, such as functional form of splines, etc. Were the defaults from the mgcv package used? If important inference is drawn from the GAMs, it would be good to assess the sensitivity of the results to choices made in the GAM fitting process. As written, it is not reproducible.

> Indeed, we used the defaults from the mgcv package, which one exception. The 'gamma' statement of the gam() function has been increased slightly to increase the degree of smoothing (to result in a smoother fitted line). This, however, did not (or only marginally) influence the inference and conclusions drawn from the model, i.e. P values for smooth terms reported in the main text and Tables S4 and S5 were highly

**BGD**

significant in either case. To improve clarity, we provide some more information on the GAM specification (including the type of splines) in the Supporting Information.

? Line 277: Because all columns of Table 4, besides the two leftmost, are in the same units (mean and SE of aboveground biomass yield), it might be better to convey the information in this table with a figure. Currently it is difficult to visually extract the most salient patterns from the table. If you do not want to use a figure maybe another possibility would be to use colors or cell fills to show where the highest values in each year were recorded.

> We actually visualized the response at the different CS, but for clarity reasons we did not divide the data by years. Please compare figure 1. Plotted over the altitude of the respective CS it show yields, aggregated by N deposition treatment (Fig 1A) and by irrigation treatment (Fig. 1B). Figure 2 shows productivity response per CS and per irrigation treatment over degree days (Fig. 2A) and the soil moisture (Fig. 2B) proxy. We believe that four panels displaying the productivity of the different CS is already a lot. Therefore, we opted to use a table for the year by year information.

? Line 281: I am confused why -7.7% is described as an increase, is it a negative or positive change?

> When viewed on the computer monitor l. 281 always says +7.7 % ('plus7.7 %'), not -7.7%, in all versions of the document. Thus 'increase'. Maybe there is a .pdf > printer problem, in case you worked on a printed copy?

? Line 289: Refer to the statistical test result (I am assuming this is Table A2?) that supports the statement that there was no significant interaction between N treatment and CS or irrigation.

> Thanks for drawing attention to this. We added the reference. It is actually Table 3, as referred to in the previous line, where the non-significant single factor N treatment is reported.

? Line 316: "climate scenario warming" is a confusing phrase. Do you mean warming consistent with some particular climate scenario?

> We tried to improve clarity and the sentence now reads 'We found a substantial and significant positive effect of climate scenarios, equivalent to warming of up to + 1.8 °C (Apr. – Oct. mean) on aboveground biomass of subalpine grasslands (up to +44 % yield).'

? Line 390: I found this paragraph to be a little confusing. Are you referring to results from the present study or previous studies in the literature? Also, because you mention specific species responses to N addition from other species, it would be more interesting if you would draw a more direct connection with the present study. Were there any individual species that you can point to their responses?

> Sorry for being unclear. We improved the wording of the two paragraphs concerned. With the paragraph you mention, our discussion of N-effects in previous studies is contrasting single key species responses with whole plant community responses, discussed in the previous paragraph. In the next paragraph a direct connection with the present study and specifically with the response of Carex ssp. is made, to explain the lack of an N-response in our experiment.

? Line 425: I am not sure what the grounds are for stating that subalpine grassland productivity will increase with warming. Is it necessarily the case that climatic conditions will "move up" in elevation – maybe there will be novel and unpredictable combinations of temperature and moisture not tested here.

> We state that subalpine grassland productivity will increase with warming, because we found that yields in our experiment were increasing with increasing climate scenario mean temperatures (warming). The future climate may actually show those 'novel and unpredictable combinations . . . ' you mention. But even though there is no way to prove it, we indeed assumed that climatic conditions will 'move up' under global warming conditions. The reason is that we considered it the most conservative choice, to keep

the number of necessary assumptions as low as possible.

---

## Author Comment (AC4) · 17 Nov 2020

Below, please find the authors point by point replies. For ease of reading we quote the comments with a '?' first and start our responses with '>'

? General Comments In their manuscript titled "The rising productivity of alpine grassland under warming, drought, and, N-deposition treatments", the others describe a novel experiment in which monoliths of soil and turf were transplanted across an elevational gradient combine with fertilization and water addition treatments. After four years of growth in the transplanted location, the others describe how plant productivity in the monoliths responded to the interaction of different temperatures (comparing climate at the transplant location to the original site where the turfs were harvested

from), fertilization, and increased moisture, as well as the interactive effects of these three treatments. The results of this study showed that intermediate levels of warming increased plant productivity, even in drier conditions. Increasing the precipitation received by some monoliths had only marginal effects on plant productivity, while fertilizing the plots with nitrogen solutions had no discernable effect on plant productivity. While this experiment is truly novel in its use of monolith transplants to simulate climate change in conjunction with two additional global change treatments in order to understand how multiple facets of global change will impact productivity, I have several concerns regarding the framing of these treatments, the metrics used to communicate and aggregate results, and the overall clarity of the manuscript. In particular, while transplanting monoliths to new elevations does of course impact climate, and in some cases results in warming, characterizing this experiment as a "warming experiment" is disingenuous. I encourage the authors to refer to their experiment as is, a transplant experiment across an elevational gradient. Furthermore, it is also a misnomer to refer to the precipitation manipulation component of this experiment as a "drought treatment", as water was added to some monoliths instead of removing precipitation, as when using rain-out shelters etc., to simulate drought.

> Indeed, in the headline we imply that we have a warming treatment, even though what we apply is an altitudinal transplantation treatment. Analogously, drought, as a productivity limiting factor, is not a treatment, but also a consequence of the downward transplantation in our experiment. We chose this wording to quickly convey motivation and relevant drivers the experiment. The supplementary precipitation (not mentioned in the headline) is a treatment to mitigate drought conditions and addressed appropriately in the text.

? My detailed line comments below elaborate on these concerns as well as my suggestions and critique of the metrics that the authors chose to describe climate in this study.

? Line Comments 34–"... to have beneficial effects": Beneficial effects on what?

> Sorry to be unclear. Old sentence reads 'First, mitigation of the thermal growth limitation is likely to have beneficial effects.' New sentence reads 'First, mitigation of the thermal growth limitation is likely to have beneficial effects on productivity.'

? 35-36: Clarify what you mean by "initial water supply"... Water resources at the beginning of the growing season are generally plentiful? But this would be the case only for plants that emerge early in the growing season, i.e. depends on phenology of plant species.

> Yes, water resources at the beginning of the season are generally plentiful. As we state behind the comma '..., because even a small winter snowpack supplies a large soil moisture resource in spring.' Plants in subalpine grasslands are all perennial and usually start greening even before the snow-cover has completely disappeared.

? 38–"kg N ha-1 a-1": These units are unconventional, instead of a-1 (per annum?) I typically see yr-1 when describing nitrogen deposition rates.

> Yes, 'per annum'. Not unconventional. The SI convention for English year is 'a'

? 45–"...showed a twofold productivity increase": In response to what treatment?

> In response to increased summer temperature. The whole sentence in the manuscript reads 'For example, tundra vegetation showed a twofold productivity increase, driven by increased summer temperature (Van der Wal and Stien, 2014)'. For the revised MS we will complement ' ... up to twofold ...' to better reflect the quoted authors statement.

? 47–"...grasses were favored over forbs and sedges by drought and warmth": This seems unclear, what do you mean by "favored by drought and warmth"? Productivity of forbs and sedges increases with warming and drought?

> Sorry, unclear. By 'favored' we mean that grass relative abundance increased at the expense of sedges and forbs. New sentence reads 'In contrast, Liu et al. (2018) combined long-term observations with a manipulative experiment to find that total net

primary productivity (NPP) in Tibetan grassland remained unaffected, though relative abundance of grasses was increased at the expense of forbs and sedges by drought and warmth.'

? 61–"...if only a short or linear segment out of a larger range of biologically possible responses is represented in the data.": There is some indication that productivity relationships revealed in manipulative experiments actually encompasses even more variation than occurs naturally (see Jochum et al. 2020. Nature Ecology and Evolution).

> Here we are making a point to include many factors and factor levels in climate change experiments, in order to avoid wrong interpretations when interpolating between data points. The original sentence reads 'Not only can a low number of treatment factors, but also a low number of treatment levels invite overly simplistic interpretation of experimental results, if only a short or linear segment out of a larger range of biologically possible responses is represented in the data.'

We do not understand how this is related to your comment above. With respect to biodiversity experiments the Jochum et al. paper finds that biodiversity experiments 'have greater variance in their compositional features than their real-world counterparts'. But based on their analysis they later conclude that this does not impair the applicability of the results of biodiversity experiments: ' ... our results demonstrate that the results of biodiversity experiments are largely insensitive to the exclusion of unrealistic communities and that the conclusions drawn from biodiversity experiments are generally robust.'

? 67–I think that I am still confused by what you mean by "factor levels"... Does this refer to consideration of multiple global change factors, or does it refer to the magnitude of the global change treatment imposed by the experiments?

> Indeed, factor levels, as in our sentence '... the outcome ... depends to a large degree on the chosen factor levels ...' refers to the magnitude of the chosen global

change treatment factor, e.g. an N-deposition with levels of 0 (control), 3 and 15 kg N ha-1 a-1 on top of the background deposition.

? 68– "Here, we present four-years of treatment results from a field experiment in the Swiss Alps.": This statement is an important introduction of your experiment, and as such, you should be more descriptive than "treatment results from a field experiment". What types of treatments specifically were involved in your field experiment, and were any of these treatments applied simultaneously to study interactive effects?

> As reviewer #2 demands, the lines 68-75, following the sentence quoted in l. 68, wrap up what we did in the experiment. Only it is not in the first, introductory sentence.

? 83–"monoliths (ML)": I do not feel that it is necessary to use an acronym for one word, and stating monolith regularly instead of ML will improve the clarity of your manuscript.

> We agree that abbreviations should be used conservatively, but we are undecided about this issue. After all the term occurs 36 times across the MS.

? 102-103: This sentence is rather unclear. What do you mean by standardizing harvests and the "zero-year" and "acclimation" distinctions? This aspect of your methods deserves an elaboration.

> We recognize the cause of confusion. Indeed, the distinctions 'zero-year' and 'acclimation' are obsolete in this place. They derive from the chronology of establishing the experiment. The 'standardizing' harvests in these first two years served to homogenize the canopy of the monoliths, that were originally grazed and therefore had more heterogeneous canopies than mown grassland. New sentence: 'Standardizing harvests were done in 2012 and 2013, to homogenize the canopy of the previously grazed monoliths, that had more heterogeneous canopies than mown grassland.'

? 111-115: I find your naming convention, using the 'CS' designations, to be needlessly confusing. These are simply sites along an elevational gradient, so why not refer to them either by their numeric elevation (i.e. 2360 m) or simply as Elevation 1 (lowest

elevation), Elevation 2.... etc., instead of introducing a less intuitive naming system.

> We chose the term 'climate scenario' (CS) to make clear, that these sites are associated with a very complex treatment, containing a number of factors. Namely, the treatment includes changes in soil moisture, temperature and growing season length. When the data is presented in the text or in figures over x-axes, that designate temperature or moisture values, it is more intuitive to use names like 'climate', that commonly associated with temperature or soil moisture values. But similar to the monolith ML issue we work on avoiding the abbreviation CS.

? 116–"...6 CS, 6 MLs from each of the six sites of origin": I find your naming convention, using the 'CS' designations, to be needlessly confusing. These are simply sites along an elevational gradient, so why not refer to them either by their numeric elevation (i.e. 2360 m) or simply as Elevation 1, Elevation 2.... etc., instead of introducing a less intuitive naming system.

> Please compare the response above (l. 111-115)

? 119–"...were filled with soil to prevent air flow": Where did this soil come from? Bulk soil from each specific elevation/origin location?

> The soil used originates from the respective scenario site, i.e. from the pit that was dug to accommodate the transplanted monoliths. This means the 'filling-soil' was not the same as in the monoliths (that come from six different origins). This does not affect the individual turf monoliths soil properties, because the monoliths remained in their drained containers for the whole duration of the experiment, so that the monolith-soil was isolated both from neighboring monoliths with different soil and from the 'filling-soil'.

? 121–"cross-factorial design": Full-factorial design? I'm unfamiliar with "cross-factorial" experimental designs.

> Yes, thanks, we got that wrong. We change that to 'full factorial'

? 153: This sentence is rather unclear... Temperatures were summed across one day?

> Unfortunately we can't find a reference to temperatures in l. 153. Our best guess is, your comment refers to l. 148 ff, saying 'The thermal energy was expressed as degree day values (DD0°C), resulting from hourly air temperature means above a threshold of 0 °C, added for one day, then divided by 24.'

We are sorry we were not clearly describing our standard procedure to calculate degree days from hourly temperature means. Indeed, there is a plethora of 'degree days', tailored to suit many specific purposes and there is no single convention. We improved the situation and the complete section now reads: 'The available thermal energy was expressed as degree days (DD) above a threshold of 0 °C (DD0°C). To derive DD we calculated the sum of hourly temperature means above 0°C during one day, then we divided this sum by 24 hours. To quantify the total thermal energy available for growth, we summed degree days during the snow-free period between the annual harvests (DD0°Ctotal), considering that the perennial vegetation continues to grow after mowing.'

? 154-156: This threshold seems particularly arbitrary, and I think that the use of a threshold in general is not necessary here. Why not simply present the mean growing season soil volumetric water content for each site/each season? This metric is much simpler and more intuitive for readers to understand and compare your results across the elevational gradient.

> We considered using mean growing season soil volumetric water content and dismissed the idea. The reason is similar to the problems arising when using mean temperatures: The plants do not experience 'mean' water contents, when coping with environmental growth limitations. For example, when plants experience a wet month after a dry month, the mean soil moisture may suggest perfect growing conditions, when they were bad indeed. We do not think that an increasing number of dry situations is less intuitive than a decreasing number of soil water content.

? 161-162: Why does the amount of precipitation added to each monolith vary between years?

> The application of the irrigation treatment was limited by the concurrent availability of workforce and occurrence of dry soil situations. We would have preferred to add more water, but did not manage to.

? 168: Listing the chemical formula of ammonium nitrate is not necessary.

> Thanks for mentioning that.

? 226: Is there some type of relationship between atmospheric N-deposition rates and elevation? Perhaps describe N-deposition rates across the entire gradient, not just at the middle and low points of your elevational gradient.

> We only have data for the second highest site CS2reference (3.3 kg N ha-1 a-1) and the lowest site CS6 (4.3 kg N ha-1 a-1). This difference likely reflects the distance of the CS from the (agricultural) N-sources. CS6 (1680 m a.s.l.) is close to the village, CS2reference (2170 m a.s.l.) is further up the mountain.

? 236: What does non-continuous mean? Non-linear?

> We wrote 'We observed a small, non-continuous increase of precipitation with altitude during April – October. The recorded annual precipitation sum was somewhat larger than the sum for the growing period (Tab. 2).' We meant to say that precipitation was not continuously rising with altitude. The second sentence refers the reader to Tab. 2, that contains the precipitation data for all sites. Also, we did not mean 'non-linear', as we did not attempt to fit a (non-linear) model to the data.

? 239–"...only one third of the pre-harvest period was dry": It is definitely a misnomer to describe conditions of lower than 40% moisture content as "dry". In fact, in most alpine systems, 30% moisture content is considered ideal moisture conditions for optimal microbial activity (see Hawkes et al. 2017 PNAS for a relevant discussion related to respiration and soil moisture). I would highly suggest re-characterizing the way in

which you describe soil moisture in this manuscript. Instead of creating a binomial in soil moisture conditions around an arbitrary 40% moisture content threshold, why not just describe average soil moisture across the growing season on a continuous scale, i.e. just state average growing season soil moisture for the pre-harvest period.

> We agree that it would be advantageous to find a better term than plain 'dry' for sentences like this. This will be improved throughout the revised manuscript. As explained in the Material and Methods section (l. 153-156), the 40% threshold was neither chosen arbitrarily nor do we imply values < 40 % to cause drought stress. Instead, the threshold was found empirically to provide a good contrast between CS and years. It is also clear, that the soil water availability is decreased in periods with an increased percentage of days with SWC < 40%. We find the Hawkes et al. 2017 paper brilliantly studies the legacy of local climatic history on differential, local microbial adaptation. They find that microbial respiration is effectively locally specialized to soil moisture conditions. We could not find references to plants, plant productivity, ideal moisture conditions or alpine sites. We do not agree with the idea to describe the water related growing conditions as 'average soil moisture across the growing season' and reiterate our response to a comment above (154-156:): 'We considered using mean growing season soil volumetric water content and dismissed the idea. The reason is similar to the problems arising when using mean temperatures: The plants do not experience mean water contents, when coping with environmental growth limitations. For example, when plants experience a wet month after a dry month, the mean soil moisture may suggest perfect growing conditions, when they were bad indeed.'

? 248-249: Because you describe soil moisture conditions in the previous section using percent dry days, we have no way of understanding how this transplantation effect on soil moisture conditions (described using VWC) might interact with your other treatments.

> Yes, there is a way of understanding the transplantation effect. In the section quoted, we state both the SWC for transplanted monoliths and the undisturbed grassland in

the simplifying 'average SWC' metric. In any case, the good message will be clear, because the difference stated is only 1% vol SWC.

? 251: I would suggest that productivity is the more appropriate term, consistent with literature in this area of ecological research, to describe your response variable.

> We will change that to 'productivity'. Unfortunately we can only offer a crude proxy for 'productivity' (net ecosystem productivity). The harvestable part of the canopy is less than net primary production. Also, due to the fixed cutting height, this metric creates an overestimate of positive effect sizes, that is larger when the yields are small. For the above reasons we replace the unspecific term 'productivity' with 'yield' if we can.

? 259: In order to show evidence to support this claim, I would like to see a figure and the related statistics that shows the relationship between the productivity effect size (productivity in transplanted monoliths - productivity in control monoliths that were reinstalled at the same site / standard deviation of productivity across all monoliths) regressed against the temperature difference from the monolith's original climate and the transplanted climate. In other words, how much of the change in productivity is explained by change in temperature following transplantation?

> Strictly speaking, we claim ' ... we found a highly significant effect of the CS ... ' in l. 259. Here, we do not claim that temperature caused the significant differences in yield. Instead, we refer to the climate scenario (CS) as a whole, because it is one of the strengths of our experiment, that we simulate climate change in the mountains as complex climate scenarios, including simultaneous changes in thermal energy, growing period length, water availability and increased pollutant deposition. An abundance of aspects of the yield response is displayed in four panels (Fig. 1 and Fig. 2) and in Tab. 4, with the statistics shown in Tab. 3 (+ Appendix Tab. A2 and A3). But, as demanded by the reviewer, we have also broken down our analysis (generalized additive models) to individual, environmental parameters of CS, namely degree days (DD0°C) and < 40% SWC conditions (dry days %). This information can be found in Results l. 299-

302 (DD0°C) and l. 303-308 (dry days %) and in Mat. & Meth. l. 215-219.

? 260-261: What does "intermediate warming" mean here? Describing this result as "monoliths that experienced X-Y degrees of warming by being transplanted to warmer climates at lower elevations relative to climate at their original location showed increases in productivity".

> 'Intermediate warming', is a term we use in the context of distinguishing the climate scenario sites (CS). It refers to those CS where the altitude related warming component of the climate change scenario is in the middle between minimum warming and maximum warming. We think that once the reader arrives at the Results section, 'intermediate warming' in the context of this study is clear enough. We aimed at keeping the Results section comprehensive, but short. Thus, we would rather not repeat the Mat. & Meth. as suggested.

? 262-264: This sentence is confusing. 2016 was the year in which productivity, on average, was highest, but this was only the case at two sites? These two statements seem to contradict one another.

> Sorry for causing confusion. Actually, we don't say that it was only the case at two sites. Only CS5 did not show maximum yield in 2016 (the corresponding numbers are in Tab. 4). Our l. 262-264 says 'In the year of the overall maximum productivity (2016), both the coldest site CS1 and the warmest site CS6 produced their respective record yield (Tab. 4).' We will replace 'both' with 'also' to be more clear. We use the term 'also' to draw the attention to a counterintuitive situation: Despite transplantation into contrasting environments (cooler at CS1 and substantially warmer at CS6), production of the maximum yield coincided with the weather conditions of the same year.

? 298: The title of this section seems to not relate to the results described within the section. You already stated that each elevational site is characterized by different temperature and precipitation regimes in your methods and in previous sections of the results. Should this section describe the relationship between productivity and climate

at each elevation?

> Very helpful! This section really describes the relationship between biomass yield and those environmental parameters (warmth and moisture) that we quantified for the individual climate scenario (CS) sites. This is different from the approach that treats CS as categories that integrate multiple climate change aspects. Accordingly, we will change the title. We suggest '3.2.5 Biomass yield response strongly relates to temperature and soil moisture changes'

? 325-326: Are there examples of other papers whose conclusions about the use of degree days instead of mean temperatures over the same time frame?

> Particularly in environments with strong temperature contrasts (day/night, summer/winter) like mountains or deserts, the use of DD does constitute a much more valuable metric for plant usable thermal energy. Similar to mean soil moisture values, mean temperatures can be extremely misleading, because a sequence of hot and freezing temperatures may well result in a comfortable average temperature that the plant has never experienced. Indeed, the whole concept of mean values is of quite limited use, just like dressing for the outdoors according to the current calendar month, instead of testing the air in front of your door. Some examples for the use of DDs in the context of grassland research are - Dukes et al. PLoS Biology 2005 (Jasper Ridge Experiment (CA)) - Fridley et al. Nature Climate Change 2016 (plant funct. strategies of 20 years UK grassland warming) - Wang et al. Ecology Letters 2020 (extremely dry Tibetan alpine grassland) - Wilsey et al. Journal of Applied Ecology 2018 (42 US grassland sites) - Zimmermann and Kienast Journal of Vegetation Science 1999 (Swiss alpine grasslands)

? 333-341: This section would benefit from a description of why the authors suspect that warming beyond "intermediate warming" was not associated with the same boost in productivity that was associated with intermediate warming.

> Good point. Originally we only implicitly described that (l. 338-341). New formulation:

'In the extreme treatment at lowest CS6 (+3 °C Apr.-Oct., +2.4 °C annual mean) the positive response to warming finally ceased to increase, but yield was still somewhat larger than at CS2reference. This demonstrates that the growth benefit from the larger thermal resource compensated for a radically smaller soil water resource (compare Figure 2 A & B). But the comparatively low growth response suggests, that the water supply at CS6 has already reached a critically low level.'

? 337–"cockchafer (Melolontha melolonth) infestation: Please describe what this organism is and how it is relevant to variability in productivity.

> The Cockchafer is a bug, its larvae feed on roots. When there are many, they may kill the vegetation. The Cockchafer (together with the Locust) is probably the one insect that is best known to the public for its periodical mass flight-years. In these years it is a major pest. We would prefer not to add too much general biology to the text.

? 347-349: Grammatical errors and diction in this sentence make it unclear.

> Reformulated sentence to be clearer: 'Also, the dramatic temperature dynamics that occurred during the past 12,000 years of the Holocene interglacial, suggest that temperature adaptations, that are still contained in modern plant genotypes, may actually match not only today's weather, but also warmer and cooler climate conditions.'

? 358: I think this statement describes my point about eliminating your use of the "percent dry days" metric entirely... Your results, using this metric, prevent readers from relating the soil moisture conditions present in your experiment to soil moisture conditions elsewhere. Furthermore, describing soil moisture conditions less than 40% as "dry" is a misnomer.

> We admit that between-experiment comparisons of soil moisture conditions, or rather the water availability for plants, is close to impossible. The reason is that A) different plants have different capacities to exploit the moisture resource. That means that a species from one experiment thrives at the same SWC when a species from another

experiment dies. B) different soils have different water potentials (osmotic plus matrix potential). As a result, soils with the same vol. % SWC may have totally different water availabilities from a plant perspective. Consequently, we chose to generate a wide range of within-experiment soil moisture conditions for comparison, rather than refer to literature values. Else, we believe that quantifying environmental conditions by describing them as more or less dry days occurring, gives the reader a perfect idea of which situation was more beneficial and which less so. As stated above (reviewers comment l. 239) we agree that 'dry days' is not a perfect choice, as dry implies a critical situation for the plant. But the metric '% days < 40% SWC' serves very well to distinguish between scenario sites, years and irrigation treatments. This highlights its value for describing the situation. Please cf. tables 2 and 4.

? 380: What caused increased evapotranspiration at CS5? Is it possible that too much rainfall, either ambient or added as part of your irrigation treatment, could cause leaching of important soil nutrients, with higher VWC leading to lower productivity? This might be especially relevant in monoliths that received both an irrigation and fertilization treatment.

> We considered that it was likely the higher temperatures in those climate scenario sites (CS3, CS4, CS5 and CS6), that were located downslope from our reference site CS2reference, that caused higher evapotranspiration. We have no reason to assume that there was too much rain. The nearby federal meteorology station recorded 662 mm/year during the experiment, while the 1981-2010 mean is 706 mm/year. Indeed, the Massenerhebung effect creates a continentality of the climate that makes inner-alpine valleys like the Engadin quite dry. Please also compare tab. 2. Our irrigation treatment only added 12-21% of the seasonal rainfall, the nitrogen deposition treatment was equivalent to 20 mm precipitation per year for all monoliths. This is not a likely scenario for nutrient leaching.

? 399-402: These are the only lines of this section of your discussion that reference your results directly. These sentences should be moved up in this section, and you

should eliminate the references to other experiments with results that contradict what your experiment found, as this section is very unclear as curssrently written. Which of these citations and theories help explain your results? Remove the rest.

> The references to other N-deposition experiments are carefully chosen and reflect the best comparisons available. Early experiments applied high doses to test for N-limitation in general. The deposition rate in later experiments was lowered, to approximate realistic atmospheric N-deposition rates. Our experiment approached the critical N-load, the limit between a responding and a not responding ecosystem. In the section in question, the relevant differences compared to our experiment are highlighted: Sometimes the deposition rates were substantially higher, sometimes single species responses or general plant community changes were reported. Summarizing these contrasts we conclude that 'the cold-adapted, mature and low productivity grassland either responds with a >5 year time lag, or that the N-deposition treatment was below the critical load for aboveground biomass responses.' We feel this is a reasonable line of argument. Also, we consider the Discussion to be a place to reflect on the state of science in the field in general, as opposed to collecting arguments that 'help explain' our results.

? 426–"This implies that subalpine grassland productivity has likely not increased during the past century warming": This statement is in no way supported by your results.

> We found that those monoliths that were subjected to a cooling treatment (at CS1), such that they experienced the temperature conditions of the 1920s, did not show a reduced growth compared to the climate scenario at CS2reference with 'modern' temperatures. What else should we conclude from that, if not that the last 100 years of warming did not affect plant growth yet?

---

## Author Response (AR1)

Dear Editor, dear Referees,

Here, we adapted the individual replies given earlier to fit the 'combined opinions' of referees and editor. We found that the comments helped to mend some flaws and recognize sections that were not sufficiently clear. As a result, we present a substantially improved manuscript.

Where we did not agree with the reviewers' points raised, we gave detailed explanations and arguments on our views. Please find our point by point replies below.

Please note that line numbers in the reviewers' comments refer to the original manuscript; but the line numbers in our responses refer to the revised manuscript with track changes displayed.

For the ease of reading we quoted the comments by the editor and two anonymous referees with an indent first, followed by our responses with '**A >**'

I would like to add that both reviewers expressed some concern about the frameing of the experiment as transplantation experiment, and I agree that the nature of the experiment should be clear already from the title. Please consider adding "... in an transplantation experiment along an altitudinal gradient" so something similar but more readible to the title.

**A >** Agreed. New, comprehensive title is:

'Sub-alpine grassland productivity increased with warmer and drier conditions, but not with higher N-deposition, in an altitudinal transplantation experiment

I would like to point out the data policy of Biogeosciences (https://www.biogeosciences.net/policies/data_policy.html). While it is of course not the intention to make the raw data of the experiment accessible, please consider seriously the request of reviewer #1 to make the statistical analysis as reproducible as possible

**A >** We will deposit the data necessary to reproduce the statistical analysis on DRYAD

(https://datadryad.org/stash). The relevant pieces of R code will be part of the revised

Appendix.

**Anonymous Referee #1**

**1 General comments**

First, I would like to see more information about the plant species composition of the experimental monoliths. Qualitative results can be informative too. This could be a few sentences in the methods. Photographs might also be helpful.

**A >** We added more information on species directly in the new M&M l. 166-170. But to keep the MS as lean as possible we refer to the Wüst-Galley et al. (2020), which reported in detail on functional group responses in the same experiment. Please find a comment on photographs below.

Generally I would be slightly concerned about the inference obtainable from the elevational gradient. If many factors change in a correlated fashion along the gradient, such as temperature, moisture, and historical human/grazing pressure, it is hard to tell which factor is the driver.

**A >** Indeed, in a complex system with a high number of interacting environmental factors, there is an equally high number of drivers. For this very reason, we have termed the sites at different altitudes 'Climate Scenario', which includes all what the reviewer claims. To uncouple altitudinal effects from soil moisture (and nitrogen) effects, we have set up an irrigation and N deposition treatment in a factorial design. To this respect we found that soil moisture and temperature, resulting from the Climate Scenario (CS) site at a specific altitude, both drive the plant productivity response (cf. Fig. 2 a,b), but the N deposition does not in a significant way (cf. *P* values in Tab. 3).

The management history of the sites of origin is very similar, but in concert with the strong edaphic factors, an effect on the present plant communities cannot be excluded. We regard this element of heterogeneity as an advantage, as it is a factor that supports the general applicability of our results.

I understand the limitations of the design and I don't think it's necessarily a flaw, but this is something that should be addressed more openly. This is also the case when discussing how soil moisture integrates information on both temperature and moisture; this could also be viewed as confounding the effects of temperature and moisture.

**A >** Agreed, the moisture of a Climate Scenario is not independent from the temperature. As stated above, we have termed the sites at different altitudes 'Climate Scenario' and analyzed the data accordingly. We assumed that the unavoidable temperature × moisture interaction closely resembles true climate change conditions, much better than an experimental manipulation of temperature or moisture alone would do.

Indeed, both moisture and temperature were related to productivity in a concerted way (cf.

Fig. 2, Appendix Tables A4 and A5). From these analyses, it can be deduced that moisture was the stronger determinant of productivity, while extreme conditions of either predictor restricted plant growth.

One other point I would like to raise about the inference is that the warming treatment is confounded with site of origin. For example, the communities subjected to highest warming were those that were moved from the highest elevation. Therefore it is difficult to say whether the different levels of warming, or the composition of plant and soil communities from each of the sites of origin, led to the different productivity responses.

This should be addressed as well.

**A >** There seem to be a misunderstanding of our design. We wish to clarify that, we did not create a warming treatment by transplanting from origins of different temperature to a common site of uniform temperature. Instead, all sites of origin have very similar temperatures and altitudes. In contrast, the experimental site, where the turf monoliths are transplanted, contained 6 climate scenario sites along a c. 700 m altitudinal gradient. Thus, six different climate treatments were established.

This misunderstanding has been clarified in an earlier round of comment/response.

For reproducibility, please make the code and data available in a repository so that readers can reproduce the results of the statistical analysis. This is especially important for the mixed model specification. Sometimes it is difficult for the reader to determine the exact model specification from the verbal description but it is easier if they can see the code.

**A >** We will deposit the data necessary to reproduce the statistical analysis on DRYAD

(https://datadryad.org/stash). The relevant pieces of the R-code will be part of the new

Appendix.

**2 Line-by-line comments**

Line 10: The abstract does a good job of stating the results of the study but it does not do a good job of stating the motivation, novelty, or broader significance of the study from the outset. Please revise accordingly.

**A >** Helpful point. We upgraded the Abstract such that aspects of motivation and broader significance are included.

Line 55: The claim that multifactorial experiments necessarily will improve predictions is debatable. Please expand on the reasoning behind this claim.

**A >** The paragraph following our claim (new l. 104 ff) is dealing with the interpretation of multifactorial (or multilevel) vs. unifactorial experiments. It contains seven references to support the argument.

Line 76: The hypotheses need to have a little more justification or explicit statement of the reasoning why the particular directions of the effects and interactions are expected.

For example, are there other studies that show similar effects or are the expectations derived from first principles?

**A >** Our hypotheses are not derived from first basic principles, as these are hard to gain due to the complexity of interacting climate change factors. Rather, our hypotheses relate to specific aspects that can be expected given the cited literature in the Introduction.

For example, we hypothesize that

'1) The effect of warming on plant growth would be beneficial at moderate warming levels, but detrimental at high warming levels.' (new l. 140-141).

Related to this hypothesis, the Introduction section mentions how warming at high altitude vs.

lowlands reduces temperature growth limitations, rather than causing heat stress. Studies on the warming effect in cold environments (in part with inconsistent results) are given in detail (new l. 91-99).

Line 85: It is interesting that southerly exposed slopes were chosen for the study. They tend to be drier and warmer than slopes with different aspect at the same elevation. I

would expect the plants living in these microclimates to be especially responsive to the warming treatment. Is this something worth briefly mentioning?

**A >** A uniformly southern exposure (and identical altitude), as opposed to different exposures, was chosen to minimize differences in climate (temperature, moisture, radiation)

between sites of origin. Also, in this region of the Alps the majority of southerly exposed slopes is used as summer pastures. In contrast, the more the slopes are pointing away from the sun, the more likely they are forested and not suitable for our grassland research.

Surely, one may assume that an adaptation to heat and drought has occurred in these plant communities, resulting in an improved tolerance against extreme events. This may make them more likely to outcompete plant communities from moist and cool habitats in the future.

We do not see though, why plants from warm and dry habitats should be more responsive to warming, judged on a 'productivity-increase per warming' basis.

Line 106: A picture says 1000 words. It would be great to have some photos of the environment at the study sites, either as a main-text figure or as a supplement.

**A >** That is a hint we were waiting for ☺. We are happy to show the AlpGrass site and the
landscape it is set in, and have added four photos to the Appendix.

Line 116: Similar to above, its would be nice to have a picture of the experimental
setup.
**A >** Cf. above

Line 150: Is there a justification for the threshold for growing degree days being set at
0C? The same goes for the 40% soil volumetric water content threshold.
**A >** Indeed, the many degree day baselines for crops or other individual species are usually
higher. We chose a 'generic' 0°C baseline, because - in a mountain environment - already
low amounts of thermal energy play an important role. Furthermore, lacking a single target
species to focus on, but working on multi-species communities instead, none of these
'specific ' baselines appeared more appropriate.
The SWC < 40 %-threshold does not imply plant growth limitation. Instead, it is an empirically
developed contrast for differences in the soil moisture status between the CSs and between
years. More time below the threshold simply means a 'drier period' in relative terms. This is
also described in new l. 260-263.

Line 196 (statistical analysis section): I'm not sure I understand the reasoning behind
assigning CS as a fixed effect but site of origin as a random effect. From my reading of
the methods those are the same thing. Can you please clarify this?
**A >** CS represents the climate scenario treatment along the altitudinal gradient at the
AlpGrass experimental site. It is an inherent treatment factor, similar to the irrigation and the
N-deposition treatments. Because we wish to make specific tests and statements of the
effect of the CS (or even one specific CS), this treatment has to be specified as a fixed effect.
By contrast, 'site of origin' represents the grassland sites where the monoliths were
excavated (and later transplanted to the AlpGrass experimental site). We do not investigate
the effects of the origins and do not make any tests or statements on origins, therefore this
variable is fitted as a random factor.

Line 216: Please include some details on the GAM fitting procedure, such as functional
form of splines, etc. Were the defaults from the mgcv package used? If important
inference is drawn from the GAMs, it would be good to assess the sensitivity of the
results to choices made in the GAM fitting process. As written, it is not reproducible.
**A >** Indeed, we used the defaults from the mgcv package, which one exception. The
'gamma' statement of the gam() function has been increased slightly to increase the degree of smoothing (to result in a smoother fitted line). This, however, did not (or only marginally)
influence the inference and conclusions drawn from the model, i.e. *P* values for smooth
terms reported in the main text and Tables A4 and A5 were highly significant in either case.
To improve clarity, we provide the GAM specification in the Appendix.
Line 277: Because all columns of Table 4, besides the two leftmost, are in the same
units (mean and SE of aboveground biomass yield), it might be better to convey the
information in this table with a figure. Currently it is difficult to visually extract the most
salient patterns from the table. If you do not want to use a figure maybe another
possibility would be to use colors or cell fills to show where the highest values in each
year were recorded.
**A >** We visualized the response at the different CS in several panels (Fig. 1, Fig 2.), but for
clarity reasons we did not split the data by years. Generally, we wish to put few emphasis on
the within year results, as minor changes in productivity from year to year must be expected
and are usually hard to explain. However, we are thankful for the idea of shading the cells
with highest values in Table 4, and did so accordingly.
Line 281: I am confused why -7.7% is described as an increase, is it a negative or
positive change?
**A >** When viewed on our computer monitors l. 281 always says +7.7 % ('plus7.7 %'), not -
7.7%, in all versions of the document. Thus 'increase'.
Maybe there is a *.pdf-file to printer communication problem, in case you worked on a printed
copy?
Line 289: Refer to the statistical test result (I am assuming this is Table A2?) that
supports the statement that there was no significant interaction between N treatment
and CS or irrigation.
**A >** Thanks for drawing attention to this. We added the reference (new l. 458). It is actually
Table 3, where the non-significant single factor N treatment is reported, as originally referred
to in the previous line.
Line 316: "climate scenario warming" is a confusing phrase. Do you mean warming
consistent with some particular climate scenario?
**A >** The sentence has been clarified (new l. 506-507).
Line 390: I found this paragraph to be a little confusing. Are you referring to results
from the present study or previous studies in the literature? Also, because you mention specific species responses to N addition from other species, it would be more interesting if you would draw a more direct connection with the present study. Were there any individual species that you can point to their responses?

**A >** We improved the wording of the two related paragraphs. Now it should be clear that these paragraphs discuss N-effects in previous studies regarding whole plant community responses versus single key species responses (new l. 588-621).

Line 425: I am not sure what the grounds are for stating that subalpine grassland productivity will increase with warming. Is it necessarily the case that climatic conditions will "move up" in elevation – maybe there will be novel and unpredictable combinations of temperature and moisture not tested here.

**A >** The future climate may actually show 'novel and unpredictable combinations … '.

However, we did not speculate about unpredictable combinations of the future climate. In our experiment, we combined three key factors related to plant growth and measured soil moisture and thermal energy, all of this reflecting many possible combinations of future climate change factors in our tested environment. Given these treatment combinations and related information, we found clear indication that yields were increasing with increasing climate scenario mean temperatures (warming). In that context, we indeed assumed that climatic conditions will 'move up' under global warming conditions. This, however, is a most conservative and reasonable choice and keeps the number of necessary assumptions as low as possible.

**Anonymous Referee #2**

Biogeosciences bg-2020-322: The rising productivity of alpine grassland under warming, drought, and N-deposition treatments

**General Comments**

In their manuscript titled "The rising productivity of alpine grassland under warming, drought, and, N-deposition treatments", the others describe a novel experiment in which monoliths of soil and turf were transplanted across an elevational gradient combine with fertilization and water addition treatments. After four years of growth in the transplanted location, the others describe how plant productivity in the monoliths responded to the interaction of different temperatures (comparing climate at the transplant location to the original site where the turfs were harvested from), fertilization, and increased moisture, as well as the interactive effects of these three treatments.

The results of this study showed that intermediate levels of warming increased plant productivity, even in drier conditions. Increasing the precipitation received by some monoliths had only marginal effects on plant productivity, while fertilizing the plots with nitrogen solutions had no discernable effect on plant productivity.

While this experiment is truly novel in its use of monolith transplants to simulate climate change in conjunction with two additional global change treatments in order to understand how multiple facets of global change will impact productivity, I have several concerns regarding the framing of these treatments, the metrics used to communicate and aggregate results, and the overall clarity of the manuscript. In particular, while transplanting monoliths to new elevations does of course impact climate, and in some cases results in warming, characterizing this experiment as a "warming experiment"

is disingenuous. I encourage the authors to refer to their experiment as is, a transplant experiment across an elevational gradient. Furthermore, it is also a misnomer to refer to the precipitation manipulation component of this experiment as a "drought treatment", as water was added to some monoliths instead of removing precipitation, as when using rain-out shelters etc., to simulate drought.

**A >** Indeed, in the headline we implied that we have a warming treatment, even though what we applied is an altitudinal transplantation treatment. This has been a commonly raised issue and we have altered the title accordingly.

We do not refer to the irrigation treatment as a 'drought treatment', but as 'irrigation treatment' (eg. section 2.3 in Material and Methods, new l. 265 ff). Drought, as a productivity limiting factor, is not a treatment per se in the experiment, but a consequence of the downward transplantation. The supplementary precipitation (not mentioned in the headline)

is a treatment to mitigate drought conditions. We feel that all of these aspects are appropriately addressed in the revised text.

My detailed line comments below elaborate on these concerns as well as my suggestions and critique of the metrics that the authors chose to describe climate in this study.

Line Comments

34–"... to have beneficial effects": Beneficial effects on what?

**A >** This has been improved (new l. 78-79).

35-36: Clarify what you mean by "initial water supply"... Water resources at the beginning of the growing season are generally plentiful? But this would be the case only for plants that emerge early in the growing season, i.e. depends on phenology of plant species.

**A >** Yes, water resources at the beginning of the season are generally plentiful. As we state in the following clause: 'because even a small winter snowpack supplies a large soil moisture resource in spring' (new l. 80).

Plants in subalpine grasslands are all perennial and usually start greening even before the snow-cover has completely disappeared. Within a given community, they reveal little phenological differences because the growing season is short; thus, growth and flowering peaks for the great majority of plants at the same time.

38–"kg N ha-1 a-1": These units are unconventional, instead of a-1 (per annum?) I

typically see yr-1 when describing nitrogen deposition rates.

**A >** Yes, 'per annum'. Not unconventional. The SI convention for English year is 'a'.

45–"...showed a twofold productivity increase": In response to what treatment?

**A >** The sentence has been clarified. For the revised MS we will complement ' … *up to*

twofold …' to better reflect the quoted author's statement (new l. 91-93).

47–"...grasses were favored over forbs and sedges by drought and warmth": This seems unclear, what do you mean by "favored by drought and warmth"? Productivity of forbs and sedges increases with warming and drought?

**A >** The sentence has been clarified. New sentence is more specific (new l. 93-96).

61–"...if only a short or linear segment out of a larger range of biologically possible responses is represented in the data.": There is some indication that productivity relationships revealed in manipulative experiments actually encompasses even more variation than occurs naturally (see Jochum et al. 2020. Nature Ecology and Evolution).

**A >** Here we are making a point to encourage inclusion of many factors and factor levels in climate change experiments to avoid wrong interpretations by missing treatment combinations or by interpolating between data points.

With respect to biodiversity experiments, Jochum et al. (2020) found that biodiversity experiments 'have greater variance in their compositional features than their real-world counterparts'. The authors later conclude that this does not impair the applicability of the results of biodiversity experiments: ' … our results demonstrate that the results of biodiversity experiments are largely insensitive to the exclusion of unrealistic communities and that the conclusions drawn from biodiversity experiments are generally robust' (Jochum et al. 2020).

We want to emphasize that our experiment does not study effects of or on biodiversity.

67–I think that I am still confused by what you mean by "factor levels"... Does this refer to consideration of multiple global change factors, or does it refer to the magnitude of the global change treatment imposed by the experiments?

**A >** It is the latter ("the magnitude of the global change treatment imposed by the experiments"). We have clarified that sentence.

68– "Here, we present four-years of treatment results from a field experiment in the

Swiss Alps.": This statement is an important introduction of your experiment, and as such, you should be more descriptive than "treatment results from a field experiment".

What types of treatments specifically were involved in your field experiment, and were any of these treatments applied simultaneously to study interactive effects?

**A >** The sentences of the whole paragraph have been improved ("Here, we present …"). As demanded by the reviewer, this paragraph wraps up what we did in the experiment. We are of the clear opinion that more information at the end of an Introduction is inappropriate, as the full information is given in the Materials and Methods section.

83–"monoliths (ML)": I do not feel that it is necessary to use an acronym for one word, and stating monolith regularly instead of ML will improve the clarity of your manuscript.

**A >** Agreed. We changed to "monolith" in the revised manuscript.

102-103: This sentence is rather unclear. What do you mean by standardizing harvests and the "zero-year" and "acclimation" distinctions? This aspect of your methods deserves an elaboration.

**A >** Indeed, the distinction between 'zero-year' and 'acclimation' is obsolete. It derives from the chronology of establishing the experiment. The 'standardizing' harvests in these first two years served to homogenize the canopy of the monoliths, that were originally grazed and therefore had more heterogeneous canopies than mown grassland (new l. 182-183).

111-115: I find your naming convention, using the 'CS' designations, to be needlessly confusing. These are simply sites along an elevational gradient, so why not refer to them either by their numeric elevation (i.e. 2360 m) or simply as Elevation 1 (lowest elevation), Elevation 2.... etc., instead of introducing a less intuitive naming system.

**A >** We chose the term 'climate scenario' (CS) to make clear that these sites are associated with a very complex treatment, containing a number of factors. Namely, the treatment includes changes in temperature, growing season length, and soil moisture. The index we chose (1-6) exactly follows your suggestion. We wish to keep this as it is.

116–"...6 CS, 6 MLs from each of the six sites of origin": I find your naming convention, using the 'CS' designations, to be needlessly confusing. These are simply sites along an elevational gradient, so why not refer to them either by their numeric elevation (i.e.

2360 m) or simply as Elevation 1, Elevation 2.... etc., instead of introducing a less intuitive naming system.

**A >** Please compare our response to the previous comment.

119–"...were filled with soil to prevent air flow": Where did this soil come from? Bulk soil from each specific elevation/origin location?

**A >** The soil used originates from the respective scenario site, i.e. from the pit that was dug to accommodate the transplanted monoliths. This does not affect the individual turf monoliths' soil properties, because the monoliths remained in their drained containers for the whole duration of the experiment, so that the monolith-soil was isolated both from neighboring monoliths. Because it seems that this detail is more confusing than helping, it has been deleted in the revised manuscript.

121–"cross-factorial design": Full-factorial design? I'm unfamiliar with "cross-factorial"

experimental designs.

**A >** This has been changed to 'full factorial' (new l. 216).

153: This sentence is rather unclear... Temperatures were summed across one day?

**A >** Unfortunately we can't find a reference to temperatures in l. 153. Our best guess is that
this comment refers to l. 148 ff:
'The thermal energy was expressed as degree day values (DD0°C), resulting from hourly air
temperature means above a threshold of 0 °C, added for one day, then divided by 24.'
Indeed, there is a plethora of 'degree days', tailored to suit many specific purposes and there
is no single convention. The section has been improved to increase clarity (new l. 255-259).
154-156: This threshold seems particularly arbitrary, and I think that the use of a
threshold in general is not necessary here. Why not simply present the mean growing
season soil volumetric water content for each site/each season? This metric is much
simpler and more intuitive for readers to understand and compare your results across
the elevational gradient.
**A >** We considered using mean growing season soil volumetric water content and dismissed
the idea. The reason is similar to the problems arising when using mean temperatures:
Plants do not experience 'mean' water contents, when coping with environmental growth
constraints. For example, when plants experience a wet month after a dry month, the mean
soil moisture may suggest perfect growing conditions, when they were bad the whole time
indeed.
We do not think that an increasing number for dry situations is less intuitive than a
decreasing number for soil water content.
161-162: Why does the amount of precipitation added to each monolith vary between
years?
**A >** The application of the irrigation treatment was determined by the occurrence of dry soil
situations, which varied among years. Therefore, no changes have been made to text here.
168: Listing the chemical formula of ammonium nitrate is not necessary.
**A >** This is a detail, which we prefer to keep. Other Journals like 'nature geoscience' do it.
The editor may decide on this, or the Journal's proof reading editor.
226: Is there some type of relationship between atmospheric N-deposition rates and
elevation? Perhaps describe N-deposition rates across the entire gradient, not just at
the middle and low points of your elevational gradient.
**A >** We only have data for the second highest site $CS2_{reference}$ (3.3 kg N $ha^{-1}$ $a^{-1}$) and the
lowest site CS6 (4.3 kg N $ha^{-1}$ $a^{-1}$). This difference likely reflects the distance of the CS from
the (agricultural) N-sources. CS6 (1680 m a.s.l.) is close to a village, $CS2_{reference}$ (2170 m
a.s.l.) is further up the mountain.

236: What does non-continuous mean? Non-linear?

**A >** It is non-linear. The text has been improved following the reviewer's assumption (new l.
386).

239–"...only one third of the pre-harvest period was dry": It is definitely a misnomer to
describe conditions of lower than 40% moisture content as "dry". In fact, in most alpine
systems, 30% moisture content is considered ideal moisture conditions for optimal
microbial activity (see Hawkes et al. 2017 PNAS for a relevant discussion related
to respiration and soil moisture). I would highly suggest re-characterizing the way in
which you describe soil moisture in this manuscript. Instead of creating a binomial in
soil moisture conditions around an arbitrary 40% moisture content threshold, why not
just describe average soil moisture across the growing season on a continuous scale,
i.e. just state average growing season soil moisture for the pre-harvest period.

**A >** We agree that it would be advantageous to find a better term than 'dry' for sentences like
this. We now use 'less soil moisture' and similar where appropriate.
As explained in the Materials and Methods section, the 40% threshold was neither chosen
arbitrarily nor does the SWC < 40 % threshold imply plant growth limitation. Instead, it is an
empirically developed contrast for differences in the soil moisture status between the CSs
and between years. More time below the threshold simply means a 'drier period' in relative
terms. This is also described in new l. 260-263. See also our response above to the same
issue (response to 154-156)
We find the Hawkes et al. 2017 paper brilliantly describing the legacy of local climatic history
on differential, local microbial adaptation. They find that microbial respiration is effectively
locally specialized to soil moisture conditions. We could not discover references to plants,
plant productivity, ideal moisture conditions or alpine sites.
With respect to the suggested use of average soil moisture values, please compare our
comment on this issue above.

248-249: Because you describe soil moisture conditions in the previous section using
percent dry days, we have no way of understanding how this transplantation effect on
soil moisture conditions (described using VWC) might interact with your other
treatments.

**A >** We think that there is a way of understanding the transplantation effect. In the section
quoted, we state both the SWC for transplanted monoliths and the undisturbed grassland
using 'average SWC'. It turned out that the difference was only 1% vol. SWC, meaning that
there was next to no transplantation effect on SWC.

251: I would suggest that productivity is the more appropriate term, consistent with literature in this area of ecological research, to describe your response variable.

**A >** We prefer to keep 'yield'. This type of grassland is maintained by mowing or grazing, and in an agronomic context, yield is fully understood and a correct term. Moreover, with our data we can only offer a crude proxy for 'productivity' (sensu net ecosystem productivity) because the harvestable part of the canopy is less than net ecosystem production. To avoid over- interpretation we prefer 'yield'.

259: In order to show evidence to support this claim, I would like to see a figure and the related statistics that shows the relationship between the productivity effect size (productivity in transplanted monoliths - productivity in control monoliths that were reinstalled at the same site / standard deviation of productivity across all monoliths)

regressed against the temperature difference from the monolith's original climate and the transplanted climate. In other words, how much of the change in productivity is explained by change in temperature following transplantation?

**A >** The wording is changed in the revised text to avoid misunderstanding (new l. 409). We do not claim that temperature caused the significant differences in yield. Instead, we refer to the climate scenario (CS) because it is one of the strengths of our experiment that we simulate climate change in the mountains as complex climate scenarios, including simultaneous changes in thermal energy, growing period length, water availability and increased pollutant deposition. We are of the clear opinion that the metric suggested by the reviewer would be misleading.

In addition, as demanded by the reviewer, we have also broken down our analysis to individual, environmental parameters of CS, namely degree days (DD0°C) and < 40% SWC

conditions (Fig. 2; fitted lines based on generalized additive models). The joint interpretation of both panels allows for a good assessment of possible drivers for yield changes over the 6

climate scenarios.

260-261: What does "intermediate warming" mean here? Describing this result as

"monoliths that experienced X-Y degrees of warming by being transplanted to warmer climates at lower elevations relative to climate at their original location showed increases in productivity".

**A >** The text has been changed to 'intermediate sites' to avoid confusion. In general,

'warming' refers to the "altitude-related warming component" of the CS, and the corresponding temperatures are given in Table 1. We think that – at this stage of the Results section – the term 'warming' should be clear in the context of the study. Moreover, we have added a formulation following the reviewer's suggestions to improve clarity (new l. 410).

262-264: This sentence is confusing. 2016 was the year in which productivity, on average, was highest, but this was only the case at two sites? These two statements seem to contradict one another.

**A >** Here, we don't say that it was only the case at two sites. In fact, all but one CS (CS5)

showed maximum yield in 2016 (see Tab. 4).

We replaced 'both' by 'also', to be more clear. We use the term 'also' to draw attention to a counterintuitive situation: Despite transplantation into contrasting environments (cooler at

CS1 and substantially warmer at CS6), production of the maximum yield coincided with the weather conditions of the same year.

298: The title of this section seems to not relate to the results described within the section. You already stated that each elevational site is characterized by different temperature and precipitation regimes in your methods and in previous sections of the results. Should this section describe the relationship between productivity and climate at each elevation?

**A >** Indeed, this section describes the relationship between biomass yield and those environmental parameters (thermal energy and moisture) that we quantified for the individual climate scenario (CS) sites. This is different from the approach that treats CS as categories that integrate multiple climate change aspects. Accordingly, we have changed the title to

'3.2.5 Yield at climate scenario sites strongly relates to changes in thermal energy and soil moisture

325-326: Are there examples of other papers whose conclusions about the use of degree days instead of mean temperatures over the same time frame?

**A >** Particularly in environments with strong temperature contrasts (day/night, summer/winter) like mountains or deserts, the use of DD does constitute a more valuable metric for plant usable thermal energy. Similar to mean soil moisture values, mean temperatures can be extremely misleading, because a sequence of hot and freezing temperatures may well result in a comfortable average temperature that the plants have never experienced.

Some examples for the use of DDs in the context of grassland research are

- Dukes et al., PLoS Biology 2005 (Jasper Ridge Experiment (CA))

- Fridley et al., Nature Climate Change 2016 (plant funct. strategies of 20 years UK grassland warming)

- Wang et al., Ecology Letters 2020 (extremely dry Tibetan alpine grassland)

- Wilsey et al., Journal of Applied Ecology 2018 (42 US grassland sites)

- Zimmermann and Kienast, Journal of Vegetation Science 1999 (Swiss alpine grasslands)

333-341: This section would benefit from a description of why the authors suspect that warming beyond "intermediate warming" was not associated with the same boost in productivity that was associated with intermediate warming.

**A >** Agreed. This was only implicitly described in the original text. We have re-structured the paragraph and have added a sentence stating that 'the comparatively low growth response suggests, that the water supply at CS6 has already reached a critically low level.' (new l.

531-533).

337–"cockchafer (Melolontha melolonth) infestation: Please describe what this organism is and how it is relevant to variability in productivity.

**A >** The Cockchafer is a bug; its larvae feed on roots. When there are many, they may kill the vegetation. The Cockchafer is probably best known for its periodical mass flight-years. In these years, it is a major pest. We have added some information, but would prefer not to add more general biology because it will hinder the flow of reading.

347-349: Grammatical errors and diction in this sentence make it unclear.

**A >** Reformulated sentence to be clearer (new l. 541 ff).

358: I think this statement describes my point about eliminating your use of the

"percent dry days" metric entirely... Your results, using this metric, prevent readers from relating the soil moisture conditions present in your experiment to soil moisture conditions elsewhere. Furthermore, describing soil moisture conditions less than 40%

as "dry" is a misnomer.

**A >** We admit that between-experiment comparisons of soil moisture conditions, or rather the water availability for plants, is close to impossible. The reasons are that

A) different plants have different capacities to exploit the moisture resource. That means that a species from one experiment thrives well at the same SWC when a species from another experiment does not.

B) different soils have different water potentials (osmotic plus matrix potential). As a result, soils with the same vol. % SWC may have totally different water availabilities from a plant perspective.

Consequently, we chose to generate a wide range of within-experiment soil moisture
conditions for comparison, rather than refer to literature values. Moreover, we believe that
quantifying environmental conditions by describing them as more or less dry gives the reader
a good idea of which situation was more beneficial and which was less so.
As stated above (reviewers comment l. 239) we agree that '*dry* days' is not a perfect choice.
Please see our answer there to the use of this term.

380: What caused increased evapotranspiration at CS5? Is it possible that too much
rainfall, either ambient or added as part of your irrigation treatment, could cause
leaching of important soil nutrients, with higher VWC leading to lower productivity? This
might be especially relevant in monoliths that received both an irrigation and
fertilization treatment.
**A >** We strongly assume that higher temperatures, in those climate scenario sites (CS3,
CS4, CS5 and CS6) downslope from our reference site $CS2_{reference}$, caused higher
evapotranspiration.
We have no reason to assume that there was too much rain. The nearby federal meteorology
station recorded 662 mm/year during the experiment, while the 1981-2010 mean is 706
mm/year. At the AlpGrass experimental site, geography implies a rather continental climate,
insofar as inner-alpine valleys like the Engadin are generally quite dry. Please also compare
Tab. 2.
Our irrigation treatment only added 12-21% of the seasonal rainfall, and the nitrogen
deposition treatment was equivalent to 20 mm precipitation per year for all monoliths. This is
not a likely scenario for nutrient leaching.

399-402: These are the only lines of this section of your discussion that reference your
results directly. These sentences should be moved up in this section, and you should
eliminate the references to other experiments with results that contradict what your
experiment found, as this section is very unclear as curssrently written. Which of these
citations and theories help explain your results? Remove the rest.
**A >** The respective sentence ('…we found no significant overall effect of N-deposition on
yield after five years …') has been moved up to the start of this paragraph. We agree that
this paragraph can be more concisely written. It has been boiled down to the most essential
statements.

426–"This implies that subalpine grassland productivity has likely not increased during
the past century warming": This statement is in no way supported by your results.

**A >** We found that those monoliths that were subjected to a cooling treatment (at CS1), such that they experienced the temperature conditions of the 1920s, did not show a reduced growth compared to the climate scenario at $CS2_{reference}$ with todays' temperatures. Given this data, it can reasonably be assumed that the last 100 years of warming did not affect plant growth yet.

---

## Author Response (AR2)

Please find the Authors responses to the subjects raised by Reviewer #1 and supported by the Handling Editor below. For ease of reading we omitted the points were the Reviewer indicated that an agreement was found and present only comments and responses were an issue had remained.

**Reviewer #1**

- Issues with inference: I am glad to read that the management history of the different sites does not differ. It is also better that the title and framing of the study, based on changes suggested by the other reviewer, now reflect that this is an altitudinal transplant, with all that entails, and not necessarily a warming treatment. However I strongly object to the statement: "We regard this element of heterogeneity as an advantage, as it is a factor that supports the general applicability of our results." It is at best neutral and at worst a diluting influence on the applicability of the results because it introduces unwanted confounding variation. However I think this statement was only made in the response to the reviewers and not the main text. If such a statement is in the main text I would suggest removing it.

**Authors:** We comply without problems. No such statement is in the text.

**Reviewer #1**

- Discussion of factorial experiments: This is improved but I do not quite follow the logic in lines 127-130: (These findings suggest that the outcome of a global change productivity-experiment depends to some degree on the chosen treatment levels and their interaction with the ambient climate during the experiment. Combining multiple treatments with many levels might thus improve interpretation of experimental outcomes and related climate change predictions.) Please clarify.

**Authors:** The quoted sentence (l. 77-80) stands in the context of the preceding arguments (in brackets). We clarified as follows (l. 77-80):

(Not only can a low number of treatment factors, but also a low number of treatment levels invite overly simplistic interpretation of experimental results, if only a short or linear segment out of a larger range of biologically possible responses is represented in the data. For example, a hump-shaped response curve (2-dimensional) under atmospheric N-deposition best described the properties of a soil C-sink in subalpine grassland (Volk et al., 2016). Similarly, a ridge-shaped response surface (3-dimensional), driven by temperature and precipitation during 17 experimental years, was needed to explain NPP data (Zhu et al., 2016).)

These findings demonstrate how, depending on the chosen treatment levels and their interaction with the ambient climate, the vegetation in a global change productivity-experiment may respond with increasing, as well as decreasing growth. Combining multiple treatments with many levels might thus improve interpretation of experimental outcomes and related climate change predictions.

**Reviewer #1**

- Details of GAM fitting: Thanks for this clarification. I would suggest including this additional detail in the appropriate place in the methods section, or alternatively include your verbal clarification that you gave in the response document in the appendix (in addition to the code). It cannot hurt to be explicit.

**Authors:** We added this information to the appendix (l. 682 ff):

**Generalized additive models to test for the effects of thermal energy (DD0Ctot) and percent days with less soil moisture (PercDryDays) on aboveground biomass yield.**

Please note that we used the defaults from the mgcv package, which one exception. The 'gamma' statement of the gam() function has been increased slightly to increase the degree of smoothing (to result in a smoother fitted line). This, however, did not (or only marginally) influence the inference and conclusions drawn from the model, i.e. *P* values for smooth terms reported in the main text and Tables A4 and A5 were highly significant in either case.

**Reviewer #1**

- Justification of statement that subalpine grassland productivity will increase with warming: I think the final statement of the concluding paragraph is still not fully supported by the data. I think the other reviewer made a similar point. Please reword this to be more suitable to the results you found.

**Authors:** We reworded the Conclusions paragraph (l. 444-446). Preceding arguments are in brackets, followed by the final statement mentioned by reviewer #1:

(Despite dwindling soil water content, the subalpine grassland growth increased to up to +1.8 °C warming during the growing period (corresponding to +1.3 °C annual mean), compared to present temperatures. Even at the maximum warming (corresponding to +2.4 °C annual mean) the yield was larger than at the reference site. At the same time -1.4 °C cooling during the growing period (corresponding to -1.7 °C annual mean) did not reduce plant growth.) These results suggest that the productivity of the subalpine grasslands in our study has likely not yet increased during the past century warming. But the positive response to warming treatments suggests, that despite growing soil moisture deficits, productivity will increase with continued warming in the near future.